# SPOROCYTELESS/NOZZLE cooperates with MADS-domain transcription factors to regulate an auxin-dependent network controlling Megaspore-Mother-Cell differentiation

Alex Cavalleri [1,7], Chiara Astori[1,7], Silvia Manrique [1,3,7], Greta Bruzzaniti[1,2], Cezary Smaczniak [2], Chiara Mizzotti [1], Alessandro Ruiu[1,4], Mattia Spanò[1], Andrea Movilli[1,5], Veronica Gregis[1], Xiaocai Xu[2,6], Kerstin Kaufmann [2] & Lucia Colombo [1] ✉

The formation of the female gamete is a complex developmental process that begins with the differentiation of the Megaspore Mother Cell (MMC) within the ovule. SPOROCYTELESS/NOZZLE (SPL/NZZ) is the principal regulator of the MMC formation, as mutations in the SPL/NZZ gene lead to the failure of the MMC differentiation. Nonetheless, the SPL/NZZ-dependent regulatory pathway governing the MMC development remains largely unknown. Using a multiomics approach, we identify direct SPL/NZZ targets and their downstream network. We discover that SPL/NZZ interacts with ovule-identity MADS-domain transcription factor complexes to regulate the expression of common target genes. By integrating the omics data with the analysis of either complementation or mutant lines, we describe a comprehensive regulatory mechanism in which SPL/NZZ controls the differentiation of the MMC by acting on an auxin-dependent downstream network.

In flowering plants, sporogenesis starts with the differentiation of female and male meiocytes: Megaspore Mother Cells (MMCs) within ovules and Pollen Mother Cells (PMCs) within anthers[1,2]. In *Arabidopsis thaliana*, the master regulator of meiocytes differentiation is *SPOROCYTELESS/NOZZLE*(SPL/NZZ), since mutations in the SPL/NZZ gene result in complete female and male sterility due to the absence of both the MMC and the PMC[1,2]. SPL/NZZ belongs to the SPEAR family, which consists of 5 genes[3,4] in *A. thaliana*. The SPEAR

proteins have a nuclear localisation signal at the N-terminus, the SPL motif in the middle of the protein, and an EAR motif at the C-terminus. It has been reported that, through the EAR motif, SPL/NZZ recruits co-repressors, like TOPLESS (TPL), that mediate the transcriptional repression of target genes by interacting with histone deacetylases (HDAs)[3–7].

SPL/NZZ also has an additional Myc-type helix-loop-helix domain that could mediate SPL/NZZ homodimerization[2,4]. Interestingly, SPL/

[1]Department of Biosciences, University of Milan, Milano, Italy. [2]Plant Cell and Molecular Biology, Institute of Biology, Humboldt-Universität zu Berlin, Berlin, Germany. [3]Present address: Instituto de Recursos Naturales y Agrobiología de Salamanca (IRNASA), Consejo Superior de Investigaciones Científicas (CSIC), Salamanca, Spain. [4]Present address: Department of Agricultural, Food and Environmental Sciences, University of Perugia, Perugia, Italy. [5]Present address: Department of Molecular Biology, Max Planck Institute for Biology Tübingen, Tübingen, Germany. [6]Present address: Department of Plant Reproductive Biology and Epigenetics, Max Planck Institute of Molecular Plant Physiology, Potsdam, Brandenburg, Germany. [7]These authors contributed equally: Alex Cavalleri, Chiara Astori, Silvia Manrique. ✉e-mail: lucia.colombo@unimi.it

NZZ was initially described as a MADS-like protein, since it contains a sequence of 17 amino acids similar to the MADS-domain[2].

According to earlier reports, SPL/NZZ can interact with two families of transcription factors (TFs): the YABBY (YAB) and the CINCINNATA-like TEOSINTE BRANCHED/CYCLOIDEA/PCF (TCP). SPL/NZZ interaction with these TFs mediates the transcriptional repression of their targets[4,5,8].

Despite its importance for plant reproduction, the mechanisms of SPL/NZZ transcriptional regulation, as well as its downstream network, are still poorly understood. Among the factors involved in SPL/NZZ regulation, auxin signalling was shown to be a positive regulator of SPL/NZZ expression in anthers, as demonstrated by the reduced SPL/NZZ transcript accumulation in the *taa1 tar2-2* mutant[9]. AGAMOUS (AG) is one of the few factors described to regulate SPL/NZZ expression in both ovules and stamens. Indeed, in flowers at stages 11–13[10], AG binds a CArG-box in the SPL/NZZ 3′ UTR to promote its expression in integuments and chalaza of the ovule. Here, the SPL/NZZ expression pattern is also regulated by the RdDM pathway, since SPL/NZZ is ectopically expressed in the *drm1drm2* mutant[11].

In addition to its function in MMC and PMC differentiation, SPL/NZZ is required for ovule patterning by negatively regulating the expression of factors such as *AINTEGUMENTA (ANT)*, *BELL1 (BEL1)*, *INNER-NO-OUTER (INO)*, and *PHABULOSA (PHB)*[12,13]. Indeed, besides the absence of the MMC, *spl/nzz* mutant ovules are characterised by an impairment in integuments growth and a longer funiculus[12].

Concerning the SPL/NZZ downstream network, SPL/NZZ plays a crucial role in the auxin signalling pathway, as the auxin-response reporter line *DR5v2* and the auxin efflux carrier *PIN-FORMED1 (PIN1)* exhibit decreased expression in the *spl/nzz* mutant ovule[14,15].

This study employs a multi-techniques approach, integrating Co-immunoprecipitation coupled with mass spectrometry (Co-IP/MS), Chromatin Immunoprecipitation and sequencing (ChIP-seq), and RNA sequencing (RNA-seq), to comprehensively identify direct SPL/NZZ targets and their downstream network. Our findings reveal the interaction between SPL/NZZ and ovule identity MADS-domain transcription factor complexes, thereby regulating the expression of common target genes, such as *ANT*. This activity influences the auxin signalling, which was previously suggested as a primary pathway necessary for the MMC differentiation[15]. We propose a multistep model in which MMC development relies on an SPL/NZZ-dependent auxin downstream network.

## Results

### SPL/NZZ and MADS-domain transcription factors interact to form multimeric complexes

To gain insight into the network regulated by SPL/NZZ, essential for the differentiation of the MMC, we identified potential SPL/NZZ partners during early stages of ovule development, before and at the MMC stage, by performing co-immunoprecipitation followed by mass spectrometry (Co-IP/MS). To do this, we took advantage of the *pSPL/NZZ::SPL/NZZ:GFP* construct, previously used to complement the *spl-1* mutant allele[11].

As previously described[11], at ovule stage 2-I[16], SPL/NZZ-GFP accumulated in the most apical cells of the L1 layer (Supplementary Fig. 1A). From stages 2-II to stage 2-V, associated with meiosis end, the SPL/NZZ-GFP accumulation domain comprised the L1 layer of the whole nucellus (Supplementary Fig. 1B–D). At later stages, the SPL/NZZ-GFP signal decreased without changing its pattern (Supplementary Fig. 1E, F).

We introduced the *pSPL/NZZ::SPL/NZZ:GFP* construct into the *pAP1::AP1:GR ap1cal* system for synchronised induction of flower development[17,18]. We performed DEX treatment on the inflorescences and evaluated the SPL/NZZ-GFP expression at different days after induction (DAI), to select samples prior to and at the MMC stage for collection. At 3 DAI, SPL/NZZ-GFP was mainly visible in another

primordia, a result in line with SPL/NZZ activity in anther morphogenesis[2,19] (Supplementary Fig. 1G–I). At 9 DAI, SPL/NZZ-GFP expression in anthers was almost absent (Supplementary Fig. 1J). By contrast, despite the partial loss of synchronisation at this later time point after AP1-GR induction, most flowers showed ovules at stage 2-I, where we could clearly observe a strong and specific nucellar SPL/NZZ-GFP accumulation (Supplementary Fig. 1K, L).

Therefore, we used three independent biological replicates of inflorescences at 9 DAI for Co-IP/MS, employing an anti-GFP antibody[20]. We found 660 proteins that were statistically enriched (log$_2$ intensity difference > 0; FDR ≤ 0.05) in the IP samples with respect to the input controls (Supplementary Data 1). As expected, SPL/NZZ itself was found as one of the most enriched proteins (Fig. 1A, and Supplementary Data 1).

To obtain an overview regarding the function of the proteins enriched in the SPL/NZZ Co-IP/MS, we performed Molecular Function and Biological Process GO terms enrichment analysis. The most enriched categories were associated with RNA polymerase activity, DNA binding and transcription (Supplementary Data 1). Moreover, we found enrichment in Biological Process categories such as floral and gynoecium development, regulation of transcription and RNA processing (Supplementary Fig. 2A, and Supplementary Data 1).

Since SPL/NZZ binds to the DNA by its association with transcriptional factor (TF) complexes[4,5], we focused our attention on the TFs identified among the SPL/NZZ partners. We found 56 TFs as putative SPL/NZZ interactors. Quite strikingly, among the TCPs, only TCP22 was statistically enriched in our list, while none of the YABBY TFs were present (Supplementary Data 1). The MADS-domain TF family was the most represented in our dataset, corresponding to almost 20% of all the TFs found (Fig. 1A, and Supplementary Data 1). This list of potential MADS-domain TFs interacting with SPL/NZZ included SEEDSTICK (STK), SHATTERPROOF2 (SHP2), SEPALLATA1 (SEP1), SEPALLATA2 (SEP2), SEPALLATA3 (SEP3) and AGAMOUS (AG), all proteins previously associated with ovule specification and development[21–23].

Based on these results, we verified whether SPL/NZZ could directly interact with MADS-domain TFs in yeast. Since SPL/NZZ contains an EAR-motif, which could negatively impact GAL4 transcriptional activation functionality, we used an SPL/NZZ variant lacking the EAR-motif sequence (SPL/NZZΔ). We initially tested the interaction between SPL/NZZΔ, fused with a GAL4 activation domain (AD) or DNA binding domain (BD) with AG, SEP3Δ[23] or STK. We found positive results for the SPL/NZZΔ$_{(AD)}$ - AG$_{(BD)}$, SPL/NZZΔ$_{(AD)}$ - STK$_{(BD)}$ and SEP3Δ$_{(AD)}$ - SPL/NZZΔ$_{(BD)}$ interactions (Fig. 1B, and Supplementary Fig. 2B). It was previously shown that MADS-domain proteins generate multimeric complexes to regulate target genes[22,24]. Based on this, we performed yeast-three-hybrid assays. We initially evaluated the formation of complexes using SPL/NZZΔ fused to the AD, SEP3Δ to the TFT and either STK or AG to the BD, finding positive results for the SPL/NZZΔ$_{(AD)}$ - SEP3Δ$_{(TFT)}$ - STK$_{(BD)}$ and SPL/NZZΔ$_{(AD)}$ - SEP3Δ$_{(TFT)}$ - AG$_{(BD)}$ interactions (Fig. 1C). It is worth noting that, even though SPL/NZZΔ$_{(AD)}$ did not directly interact with SEP3Δ$_{(BD)}$ (Supplementary Fig. 2B) and only modestly with STK (Fig. 1B), the presence of all three components enhanced yeast growth, suggesting the formation of a more stable complex (Fig. 1C). We then tested the same interactions by fusing SPL/NZZΔ with the GAL4 BD, reconfirming the establishment of complexes with either STK and SEP3Δ or AG and SEP3Δ (Supplementary Fig. 2C). Concerning the screening with SPL/NZZΔ fused with the TFT, we found that the presence of SPL/NZZΔ$_{(TFT)}$ did not enhance the already described SEP3-AG interaction[22] (Supplementary Fig. 2D).

As we confirmed that SPL/NZZ can interact with the ovule identity MADS-domain TFs, we analysed STK, AG and SEP3 expression in the nucellus at stages 2-I, 2-II, concurrently with the MMC differentiation. STK and SEP3 were expressed in the nucellus, as shown by the study of GFP-fusion-protein reporter lines, with a pattern comparable to that of

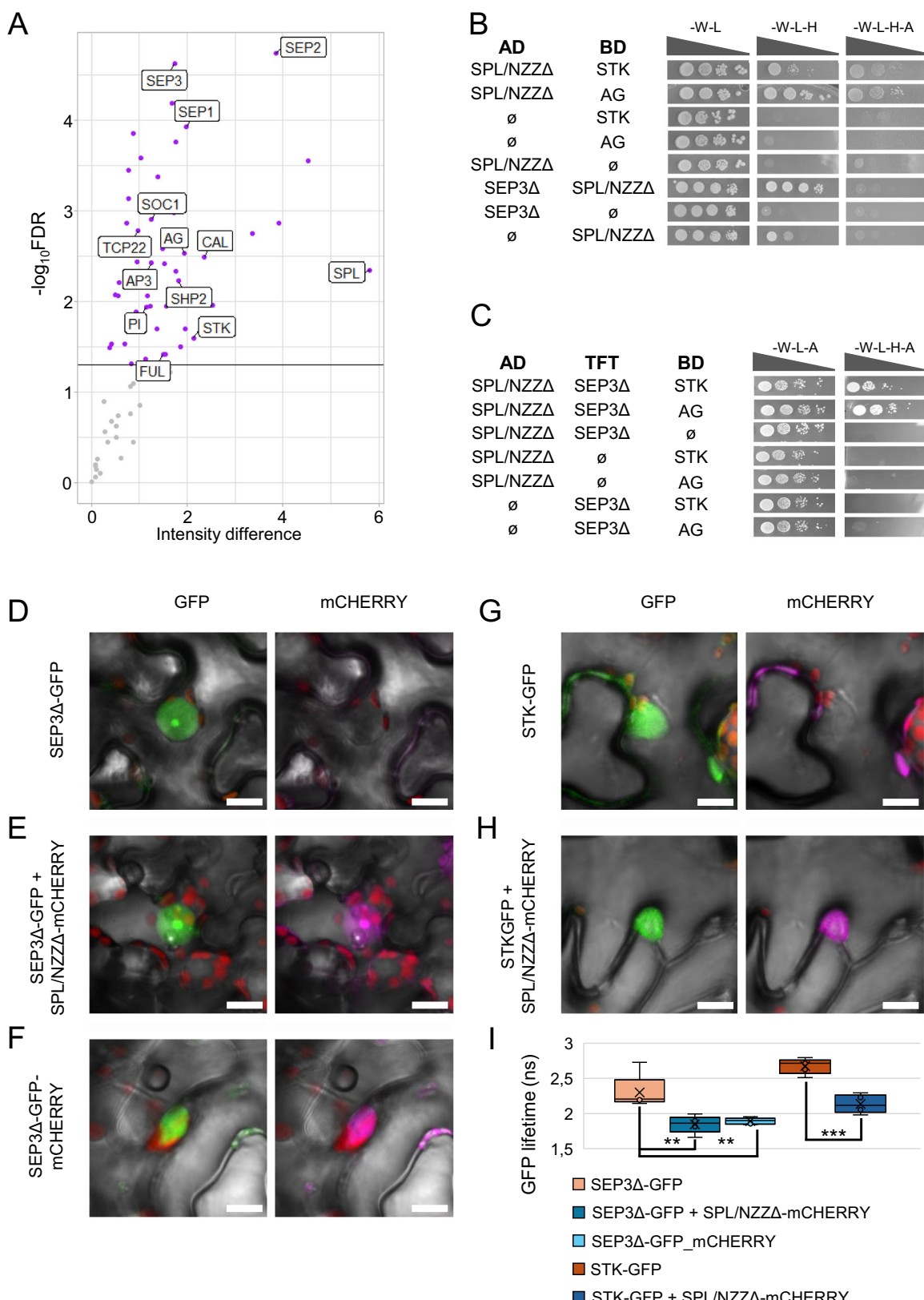

SPL/NZZ[25–27] (Supplementary Fig. 2E–H). By contrast, AG was not expressed in the nucellus (Supplementary Fig. 2I, J)[27]. To further corroborate the SPL/NZZ direct interaction with SEP3 and STK, we performed Förster Resonance Energy Transfer-fluorescence lifetime imaging (FRET-FLIM)[28] assay *in planta*. We transiently expressed, in tobacco leaves, SEP3Δ-GFP or STK-GFP fusion proteins either alone or in association with SPL/NZZΔ-mCHERRY (Fig. 1D–I, and Supplementary Fig. 3A–E). By imaging nuclei expressing solely SEP3Δ-GFP (Fig. 1D) or STK-GFP (Fig. 1G), we could measure, through FLIM, a GFP lifetime of around 2.3 and 2.7 ns, respectively (Fig. 1I, and Supplementary Fig. 3A, D). When SEP3Δ-GFP or STK-GFP were co-expressed with SPL/NZZΔ-mCHERRY (Fig. 1E, H), we measured a statistically significant

**Fig. 1 | SPL/NZZ directly interacts with MADS-domain TFs. A** Vulcano plot showing, with purple dots, the TFs statistically enriched in the SPL/NZZ Co-IP/MS dataset. MADS-domain TFs, SPL/NZZ and TCP22 have been highlighted with labels. The experiment has been performed using three independent biological replicates of *SPL/NZZ:GFP AP1:GR ap1cal* inflorescences at 9 DAI. Proteins with a positive enrichment (log₂ Intensity difference > 0) and FDR ≤ 0.05 were considered significant. The full list of SPL/NZZ interactors identified by Co-IP/MS can be found in Supplementary Data 1. **B** SPL/NZZ-MADS-domain TFs (AG, STK and SEP3) direct interaction by yeast-two-hybrid assay. Interactions are tested on media depleted of histidine (-W -L -H), or histidine and adenine (-W -L -H -A). For each interaction on the different media, yeast has been spotted at four different concentrations according to a serial dilution (OD 0.5; 1:10; 1:100; 1:1000). **C** SPL/NZZ form complexes with STK, SEP3 and AG by yeast-three-hybrid assays. Interactions are tested on a medium depleted of histidine (-W -L -H -A). For each interaction, yeast has been spotted at four different concentrations according to a serial dilution (OD 0.5; 1:10; 1:100; 1:1000). **D–I** SPL/NZZ interaction with SEP3 and STK tested by FRET-FLIM. **D** GFP fluorescence imaged from a nucleus expressing SEP3Δ-GFP. **E** GFP and mCHERRY fluorescence imaged from a nucleus expressing SEP3Δ-GFP+SPLΔ-

mCHERRY. **F** GFP and mCHERRY fluorescence imaged from a nucleus expressing SEP3Δ-GFP-mCHERRY. **G** GFP fluorescence imaged from a nucleus expressing STK-GFP. **H** GFP and mCHERRY fluorescence imaged from a nucleus expressing STK-GFP+SPLΔ-mCHERRY. **I** Box plot representing the GFP lifetime of nuclei expressing SEP3Δ-GFP, SEP3Δ-GFP+SPL/NZZΔ-mCHERRY, SEP3Δ-GFP-mCHERRY, STK-GFP and STK-GFP+SPL/NZZΔ-mCHERRY, as calculated by FILM combined with a phasor plot to distinguish different lifetime populations. Measurements have been performed on 5, 5, 5, 11 and 5 different nuclei expressing SEP3Δ-GFP, SEP3Δ-GFP+SPL/NZZΔ-mCHERRY, SEP3Δ-GFP-mCHERRY, STK-GFP and STK-GFP+SPL/NZZΔ-mCHERRY, respectively. Source data are provided as a Source Data file. Asterisks over boxes represent the statistical significance as determined by student's *t*-test, two-sided distribution, homoscedastic, confronting SEP3Δ-GFP with SEP3Δ-GFP+SPL/NZZΔ-mCHERRY and SEP3Δ-GFP-mCHERRY, or STK-GFP with STK-GFP+SPL/NZZΔ-mCHERRY. $^{**}p ≤ 0.01$; $^{***}p ≤ 0.001$. Exact *P* values for each comparison can be found in the Source Data file. Box-plots elements correspond to: centre line = median; X = average; box limits = interquartile range; whiskers = lowest and highest values in the 1.5 interquartile range. Single measures are represented with dots in the boxes. Scale Bars: 10 µm.

reduction in the GFP lifetime because of FRET occurring between GFP and mCHERRY (Fig. 1I, and Supplementary Fig. 3B, E). Since FRET could occur only when donor and acceptor proteins are at a distance lower than 10 nm, this experiment confirms SEP3 and STK ability to physically interact with SPL/NZZ in plant cells.

It was already reported that STK-SEP3 can form a tetramer[29] while SPL/NZZ can form a homodimer[4]. Using AlphaFold3[30], we predicted in silico the possible formation of SPL/NZZ-SEPs-STK complexes containing, respectively, two copies of SEP (1, 2, or 3), STK and SPL/NZZ. AlphaFold3 preliminary predictions could fold SPL/NZZ only from amino acid 61 to amino acids 91, forming an alpha helix, a protein portion that contains the 8 amino acids corresponding to the SPL-motif. Indeed, the SPL-motif was originally predicted to be part of an alpha helix spanning from amino acids 64 to amino acid 80, which has some homology to the first helix of MADS-domain TFs[2]. For these reasons, we considered only the SPL/NZZ region spanning from amino acid 61 to 91 for further predictions. All the different simulations predicted the formation of complexes with good confidence levels, assessing the reliability of the results obtained (Supplementary Fig. 4A). Regarding the docking features of SPL/NZZ to the MADS-domain TFs tetramer, the different simulations indicated that SPL/NZZ contacts the K-box of STK and SEPs (Supplementary Fig. 4B). Finally, we performed binding affinity analyses[31–33], retrieving Gibbs free energy (ΔG) values and dissociation constants (K_d) that confirm the thermodynamic stability of the SPL/NZZ-MADS complexes (Supplementary Fig. 4C).

## SPL/NZZ and the MADS-domain TFs share common targets

To identify the direct targets of SPL/NZZ, we used the *pSPL/NZZ::SPL/NZZ:GFP AP1:GR ap1cal* line to perform a ChIP-seq experiment. Three biological replicates of inflorescence at 9 DAI were used for chromatin immunoprecipitation using an anti-GFP antibody[34,35]. After input and IP samples sequencing, peaks were considered significant with a fold enrichment (FE) > 0, FDR ≤ 0.05 and if present in at least 2 out of 3 biological replicates. This approach retrieved a list of 203 peaks, corresponding to 197 different genes (Supplementary Data 2). Around 80% of the peaks mapped in genomic regulatory regions, such as promoters, supporting the reliability of the results obtained (Fig. 2A). Among SPL/NZZ-bound genes, we found genes previously linked to SPL/NZZ function as *ANT*[12], *YUCCA6 (YUC6)*[36] and *PHB*[37] (Supplementary Data 2). Interestingly, also SPL/NZZ itself was present in the list, indicating the possible existence of a feedback loop involved in SPL/NZZ regulation.

SPL/NZZ does not have a DNA-binding domain. This implies that SPL/NZZ binds to its direct targets via TF complexes in which it is

recruited. To characterise which TFs mostly relate to SPL/NZZ activity, we performed a TF DNA-motif enrichment analysis. We used the MEME suite[38] to analyse the 203 peak sequences retrieved from the SPL/NZZ ChIP-seq experiment. The only statistically enriched TF DNA binding site identified was the CArG-box consensus sequence, to which the MADS-domain TFs bind (Fig. 2B)[39]. In addition, enrichment analysis against the Transcriptional Factor Target lists from both the AGRIS server[40–42] and the PlantGSAD database[43] revealed that SPL/NZZ targets were enriched in genes targeted by MADS-domain TFs such as AG and SEP3 or floral homeotic factors such as APETALA2 (AP2) (Supplementary Data 2).

If SPL/NZZ binding to the DNA largely depends on its association with MADS-domain TF complexes, not only SPL/NZZ and MADS-domain TFs should target the same genes, but they should bind these genes in the same genomic sequences. Based on this assumption, we determined the overlap of peaks obtained by the SPL/NZZ ChIP-seq with the ones obtained from a SEP3 ChIP-seq dataset[44]. The already available SEP3 ChIP-seq was performed using the same induction system and tissue used for the SPL/NZZ ChIP-seq and on a similar stage of development (8 DAI). Over 90% of SPL/NZZ peaks (185 out of 203, corresponding to 182 different genes) overlapped with peaks in the SEP3 dataset, strongly supporting the cooperation between SPL/NZZ and SEP3 for binding to the DNA, at least at the developmental stages used for the ChIP-seq experiments. (Fig. 2C, D, and Supplementary Data 2).

Our data indicate that SPL/NZZ could cooperate with the MADS-domain TFs SEP1, 2, and 3, as well as STK and SHPs for MMC development. This outcome was also predictable, as it has been shown that MADS-domain TFs of classes C, D and E form multimeric complexes to establish ovule domains and define cell identity[23]. Regrettably, these transcription factors exhibit significant redundancy and multiple mutants, as the *shp1 shp2 stk ag/+* one, displays ovules homeotically changed into leaves[21–23], whereas the *sep1 sep2 sep3* mutant has all floral organs homeotically changed into sepals[45]. Therefore, to investigate the potential relevance of these MADS-domain TFs in MMC differentiation, we created a *pSPL/NZZ::MADSas* construct to operate the post-transcriptional gene silencing of *SEP* genes in the nucellus (Supplementary Fig. 5A). We subsequently introduced this construct into *shp1/shp1 shp2/shp2 stk/STK* plants, intending to inhibit the activity of many ovule-identity MADS-domain TFs within the nucellus. We obtained three independent transformant lines expressing the *pSPL/NZZ::MADSas* (Supplementary Fig. 5B). Specifically, lines 1 and 2 resulted *shp1 shp2 STK/STK*, whereas line 3 was classified as a triple *shp1 shp2 stk* mutant. Although the phenotype was much less penetrant with respect to that reported in *spl-1*, the transgenic plants *shp1 shp2 stk*

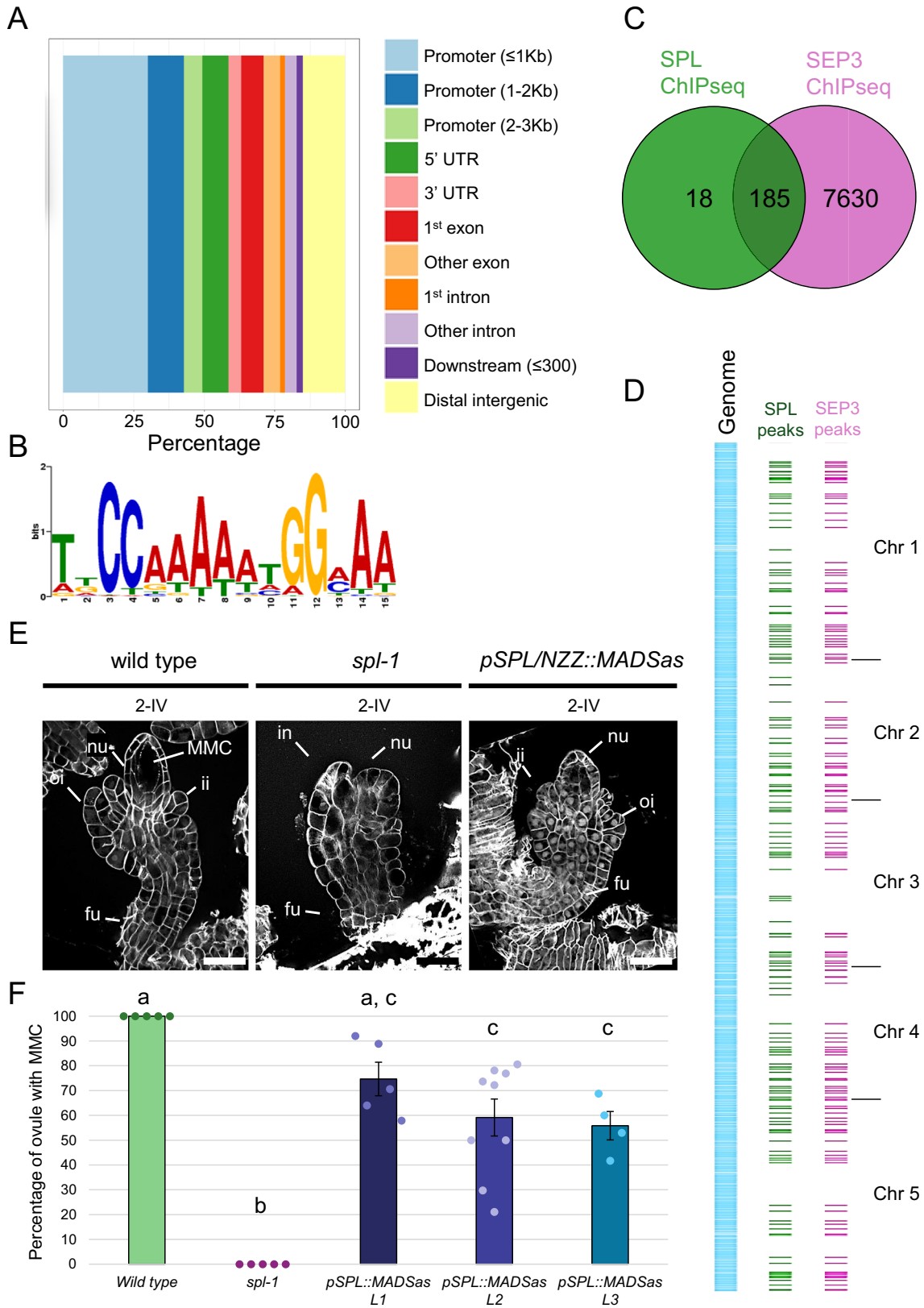

**Nature Communications** | (2026)17:683

*pSPL/NZZ::MADSas* and *shp1 shp2 STK/STK pSPL/NZZ::MADSas* showed a ratio of 25% to 45% of ovules that failed to produce the MMC (Fig. 2E, F, and Supplementary Fig. 5C).

Altogether, these data support that SPL/NZZ and MADS-domain TFs cooperate, forming a multimeric complex for the differentiation of the MMC.

## SPL/NZZ regulates the auxin signalling network

Having identified potential direct SPL/NZZ-SEP3 target genes, we focused on their biological function to determine the mechanisms involved in MMC differentiation. To do so, we performed a Biological Process GO terms enrichment analysis using the SPL/NZZ-SEP3 common targets (Supplementary Fig. 6A, and Supplementary Data 2).

**Fig. 2 | SPL/NZZ direct target genes are shared with MADS-domain TFs. A** Bar plot showing the SPL/NZZ ChIP-seq peaks localisation along the genome. Around 80% of the identified peaks fall within transcriptional regulatory sequences, such as promoters, 5′ UTRs, first exons, first introns, and 3′ UTRs. The experiment has been performed on three biological replicates of *SPL/NZZ:GFP AP1:GR ap1cal* inflorescences at 9 DAI. Peaks were considered significant with FE > 0, FDR ≤ 0.05 and if present in at least 2 biological replicates out of 3. The full list of SPL/NZZ targets identified by ChIP-seq can be found in Supplementary Data 2. **B** The CArG-box motif is the only enriched TF DNA sequence binding-site identified among the SPL/NZZ ChIP-seq peaks. The analysis has been performed with the XTREME tool from the MEME suite. **C** Venn diagram showing the intersection between the SPL/NZZ ChIP-seq peaks and the SEP3 ChIP-seq peaks. 185 SPL/NZZ peaks, representing more than 90% of the total peaks, are overlapping with SEP3 peaks. **D** SPL/NZZ ChIP-seq and SEP3 ChIP-seq overlapping peaks visualisation along the *A. thaliana* genome. All the significant SPL/NZZ ChIP-seq peaks are visualized in green. SEP3 ChIP-seq peaks overlapping with SPL/NZZ ones are shown in magenta. The full list

of SPL/NZZ-SEP3 common targets can be found in Supplementary Data 2. **E** wild type, *spl-1* and *pSPL/NZZ::MADSas* ovules at stage 2-IV. Even if with a milder effect with respect to the *spl-1* case, *pSPL/NZZ::MADSas* fails in the differentiation of the MMC, that could be clearly identified in the wild type nucellus. **F** Bar plot showing the mean ± SEM of the percentage of ovules developing the MMC in wild type, *spl-1, pSPL/NZZ::MADSas L1* (*shp1,2,STK/STK*), *pSPL/NZZ::MADSas L2* (*shp1,2,STK/STK*) and *pSPL/NZZ::MADSas L3* (*shp1,2,stk*). The analysis has been performed on 5, 5, 5, 9 and 4 different pistils, respectively, for the wild type (122 ovules observed in total), *spl-1* (110 ovules observed in total), *pSPL/NZZ::MADSas L1* (113 ovules observed in total), *pSPL/NZZ::MADSas L2* (208 ovules observed in total), *pSPL/NZZ::MADSas L3* (67 ovules observed in total). Source data are provided as a Source Data file. Letters above the bars indicate homogenous categories with *p* ≤ 0.05 as determined by one-way ANOVA with post-hoc Tukey HSD test. Exact *P* values for each comparison can be found in the Source Data file. Single measures are represented with dots in the bars. nu nucellus, fu funiculus, ii inner integument, oi outer integument, in integument, MMC megaspore mother cell. Scale bar = 20 μm.

Besides categories associated with reproductive organs differentiation, the most enriched terms were related to auxin metabolism and biosynthesis, suggesting a direct link between SPL/NZZ and auxin (Supplementary Fig. 6A, and Supplementary Data 2).

To identify the genes whose expression was modified in the absence of SPL/NZZ, and the pathways associated with SPL/NZZ activity, we performed RNA-seq on wild-type (Col-0) and *spl-1* mutant pistils at stage 9[46,47]. Differential expression analysis retrieved a list of 6877 differentially expressed genes (DEGs) (FDR ≤ 0.05) (Supplementary Data 3).

We performed a Biological Process GO term enrichment analysis on the 3502 downregulated genes, revealing enrichment in terms associated with chromatin remodelling and floral development (Supplementary Fig. 6B, and Supplementary Data 3). By contrast, Biological Process GO term enrichment analysis done on the 3375 upregulated genes confirmed an impairment in auxin biosynthesis, metabolism and signalling (Supplementary Fig. 6C, and Supplementary Data 3). We also observed enrichment in categories involved in cell cycle regulation, cell wall modification and callose deposition, processes known to be required for the MMC differentiation[48]. To further dissect the activity of SPL/NZZ over its direct targets, we analysed the SPL/NZZ-SEP3 common target list in light of the DEGs obtained from the *spl-1* RNA-seq. We found that 77 SPL/NZZ-SEP3 targets were also differentially expressed in the *spl-1* pistil with respect to the wild type (Supplementary Data 3). Nicely, these include genes involved in ovule development, such as *ANT*[49], *LUG*[50], *AP2*[51], *CUC3*[52], *IPT1*[14], *SEP3*[22], *AGL15*[53] and *FUL*[54]. Interestingly, all the auxin biosynthetic genes present in the *spl-1* RNA-seq, including also the genes targeted directly by SPL/NZZ, were upregulated (Supplementary Data 3). Key players of auxin biosynthesis, such as *YUCCA 2, 4, 6* and *TAA1*, which are normally expressed in the ovule chalaza[55], showed enhanced expression. This result is in line with SPL/NZZ proposed activity as a transcriptional repressor, which is also supported by the reduced expression of *YUC2* and *4* in the *spl-D* dominant mutant[36]. After biosynthesis in the chalaza, auxin is polarly transported towards the primordium apex mainly by PIN1, generating a region of high signalling output at the nucellus tip[55–57]. Auxin is then directed toward the L2 layer by PIN3 and PIN1 action[1,57]. Since it was already reported that *PIN1* was drastically downregulated in *spl-1* ovules[14], as well as the auxin response[15], we decided to investigate in more detail auxin transport and response in wild-type and *spl-1* ovules to evaluate their impact on the *spl-1* phenotype.

Focusing on wild-type ovules, we observed that the PIN1-GFP signal increased in the nucellus from stage 2-I, the stage in which the differentiation of the MMC initiates, to stage 2-III, associated with the complete differentiation of the MMC (Fig. 3A, B, M). Likewise, we measured a significant increase in the nucellar *DR5v2*[58] signal passing from stage 2-I to stage 2-III (Fig. 3E, F, N). This dynamic in *PIN1* and *DR5v2* expression during MMC differentiation could suggest the

requirement of auxin transport and signalling to support the MMC identity acquisition. Indeed, even though we observed that PIN1-GFP accumulated similarly in wild-type and *spl-1* ovules at stage 2–I (Fig. 3A, C, M), at stage 2-III, PIN1-GFP signal was drastically reduced in the *spl-1* mutant (Fig. 3B, D, M), as previously shown[14,15]. Likewise, the *DR5v2* signal observed between wild-type ovules at stages 2-I and 2-III was severely reduced in the *spl-1* background (Fig. 3E–H, N). Besides PIN1, PIN3 is also involved in auxin transport in the nucellus. Despite this, the *pPIN3::PIN3:GFP*[57,59] signal was localised at the very tip of ovule primordia in *spl-1*, and no differences were observed compared to the wild type (Supplementary Fig. 7A–D).

Downregulation of *PIN1* expression and *DR5v2* response suggests that auxin is accumulated at lower levels in the *spl-1* nucellus. Indeed, by using the *R2D2* reporter line[58], we detected a higher accumulation of DII-VENUS in the *spl-1* nucellus with respect to the wild type (Fig. 3I–L, and Supplementary Fig. 7E–I).

Considering the behaviour of *PIN1* and *DR5v2*, together with the fact that auxin transport was previously suggested to be required for MMC development[15], we believed that *PIN1* downregulation could be the main cause of MMC absence in the *spl-1* mutant ovule.

**Restoring the auxin transport is sufficient to rescue the MMC differentiation in *spl-1* ovules**

Even if molecularly SPL/NZZ is a repressor of transcription, during MMC differentiation, SPL/NZZ works as a positive modulator of *PIN1* expression. This implies that SPL/NZZ activity over *PIN1* is indirect and involves, potentially, the direct targeting of negative regulators of *PIN1*. Indeed, *PIN1* was not present in the list of SPL/NZZ direct targets. Taking into consideration this information, among SPL/NZZ-SEP3 direct targets, we searched for factors which could be genetically associated with SPL/NZZ activity, and which could work as intermediates of SPL/NZZ action over *PIN1*. During ovule patterning, SPL/NZZ was shown to be required for the repression of chalazal factors such as *ANT*, *BEL1*, *INO* and *PHB* in the nucellus[12,37,49]. As mentioned above, *ANT* is a direct target of the SPL/NZZ-MADS-domain TF complex. Particularly, in both SPL/NZZ and SEP3 ChIP-seq datasets, we found a peak containing 4 CArG-box sequences at around -3.6 Kb from *ANT* TSS (Supplementary Fig. 8A). Absence of SPL/NZZ function determines *ANT* ectopic expression in the nucellus[12] (Supplementary Fig. 8B–I). Our attention was caught by *ANT* as a putative intermediary of SPL/NZZ action over *PIN1*, especially due to the previous description of the ability of *ant* mutation to rescue MC differentiation in *spl/nzz* mutant background[12]. Considering the centrality of auxin transport and response for the MMC differentiation, we speculated that *ANT* could act as a repressor of *PIN1*, contributing to *PIN1* downregulation in the *spl-1* nucellus. Even though we did not find *PIN1* among ANT direct targets[60], we observed that *PIN1* was ectopically expressed in the chalaza and funiculus of *ant.4* mutant ovules with respect to the wild

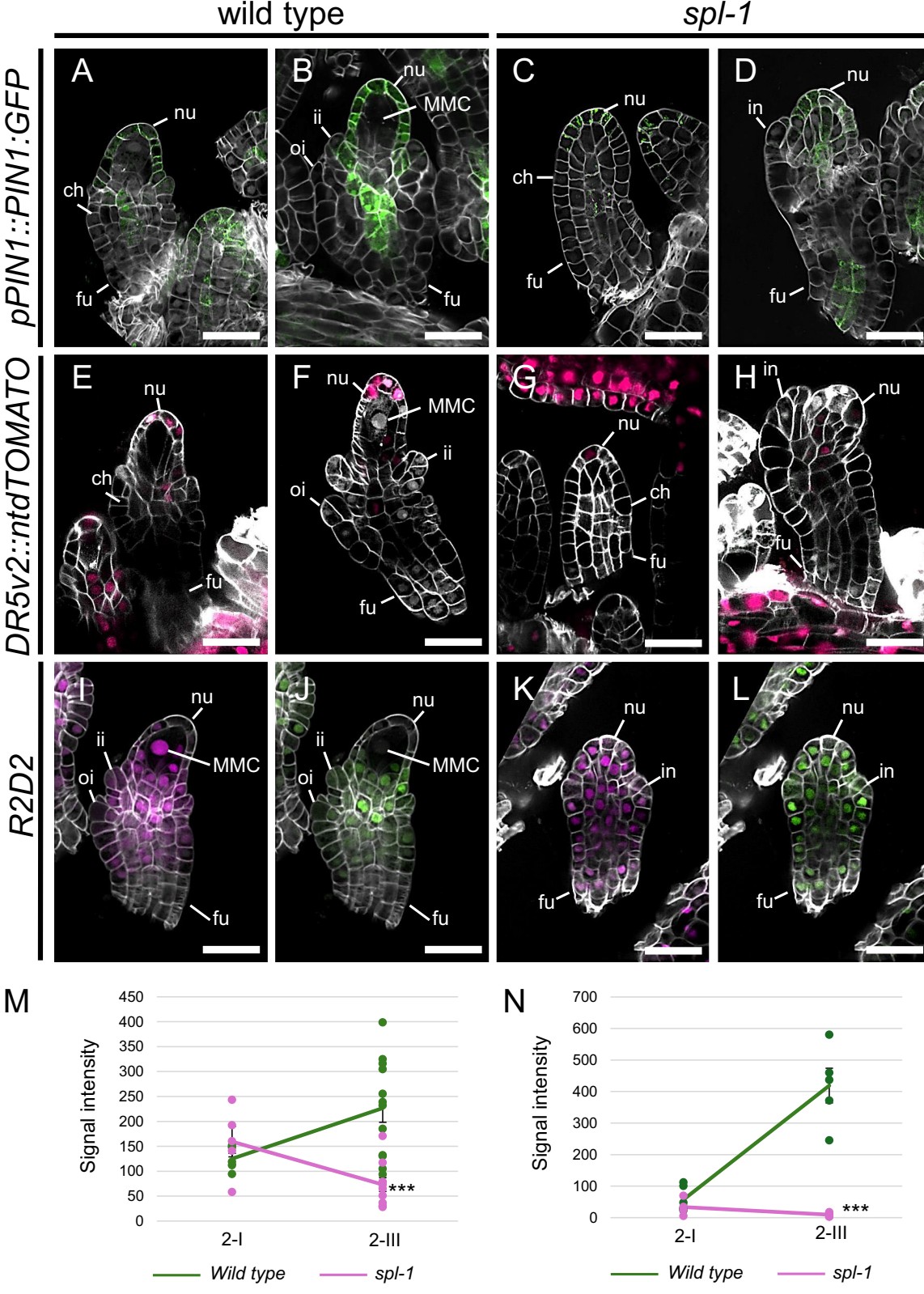

type (Supplementary Fig. 8J–Q). This suggests that, in ovules, ANT acts as an indirect local repressor of *PIN1*.

Based on our present and previously reported observations[15], we hypothesised that restoring polar auxin transport in the *spl-1* ovule might be sufficient to recover the MMC differentiation. Therefore, we developed the *pSPL/NZZ::PIN1* construct, and we transformed it into

*spl-1* mutant plants. Four distinct *spl-1* lines expressing the *pSPL/NZZ::PIN1* construct were characterised (Supplementary Fig. 9A).

In contrast with the *spl-1* situation (Fig. 4F–J, P), in *spl-1 pSPL/NZZ::PIN1*, ovule developmental defects were almost completely restored (Fig. 4A–E, K–P, and Supplementary Fig. 9B). The MMC correctly developed (Fig. 4C, D, M, N) and completed the

**Fig. 3 | Auxin transport and signalling are impaired in the *spl-1* mutant ovule.** *pPIN1::PIN1:GFP* expression in wild type (**A**, **B**) and *spl-1* (**C**, **D**) ovules at stages 2-I (**A**, **C**) and 2-III (**E**, **H**). *DRSv2* expression in wild type (**E**, **F**) and *spl-1* (**G**, **H**) ovules at stages 2-I (**E**, **G**) and 2-III (**F**, **H**). *R2D2* reporter line in wild type (**I**, **J**) and *spl-1* (**K**, **L**) ovules at stage 2-III, showing mDII-tdTOMATO (**I**, **K**) and DII-VENUS (**J**, **L**) accumulation. **I**–**L**, **M** Line graph showing the mean ± SEM of PIN1-GFP signal intensity in the L1 layer of wild type and *spl-1* ovules at stages 2-I and 2-III. Measurements have been performed on 5 wild type and 5 *spl-1* different ovules at stage 2-I and on 12 wild type and 10 *spl-1* different ovules at stage 2-III. Source data are provided as a Source Data file. Asterisks represent the statistical significance as determined by student's *t*-test, two-sided distribution, homoscedastic, confronting the mutant with the wild-type condition at each time point. \*\*\**p* ≤ 0.001. Exact *P* values for each comparison can be found in the Source Data file. Single measures are represented with dots in correspondence to the line ends. **N** Line graph showing the mean ± SEM of the tdTOMATO signal intensity, from the *DRSv2* reporter, in the L1 layer of wild-type and *spl-1* ovules at stages 2-I and 2-III. Measurements have been performed on 6 wild-type and 4 *spl-1* different ovules at stage 2-I and on 5 wild-type and 5 *spl-1* different ovules at stage 2-III. Source data are provided as a Source Data file. Asterisks represent the statistical significance as determined by student's *t*-test, two-sided distribution, homoscedastic, confronting the mutant with the wild-type condition at each time point. \*\*\**p* ≤ 0.001. Exact *P* values for each comparison can be found in the Source Data file. Single measures are represented with dots in correspondence to the line ends. nu nucellus, ch chalaza, fu funiculus, ii inner integument, oi outer integument, in integument, MMC megaspore mother cell. Scale Bars = 20 μm.

megasporogenesis (Fig. 4E, O). Indeed, fertile ovules were formed and fertilised by pollen produced by the same plants, showing that the *pSPL/NZZ:PIN1* had also rescued the PMC defects (Supplementary Fig. 9C). Interestingly, *spl-1 pSPL/NZZ:PIN1* ovules also showed a rescue of *spl-1* integuments defect (Fig. 4G–J, L–O). Indeed, ovules could correctly differentiate inner and outer integuments (Fig. 4G–J, L–O). Complementation of inner integuments development suggests that SPL/NZZ is important for the proper PIN1 localisation and auxin transport during inner integuments formation. Indeed, regulation of auxin transport was shown to be involved in integuments development[61]. Nevertheless, similarly to the *spl-1* mutant, the funiculus length in *spl-1 pSPL/NZZ::PIN1* ovules remained statistically longer with respect to the wild type (Supplementary Fig. 10A, B). We hypothesised that this phenotype could be related to the fact that genes involved in funiculus differentiation remained deregulated in *spl-1 pSPL/NZZ::PIN1* ovules. For instance, it was shown that *ANT* ectopic expression is directly associated with the differentiation of a longer funiculus[62]. By ISH, we observed that *ANT* remained ectopically expressed in both nucellus and funiculus of *spl-1 pSPL/NZZ::PIN1* ovules (Supplementary Fig. 10C). The ectopic expression of *ANT* in *spl-1 pSPL/NZZ::PIN1* not only provides a hierarchical view in which *PIN1* is placed downstream *ANT* but also represents a plausible explanation for the elongated funiculus of *spl-1 pSPL/NZZ::PIN1* ovules.

## The inhibition of ARFs activity phenocopies the *spl-1* mutation

The results described above demonstrate the need for SPL/NZZ to regulate the auxin accumulation in the nucellus, which is essential for MMC differentiation. According to the auxin response model, auxin accumulation enhances Aux/IAAs degradation, releasing the ARFs from Aux/IAAs repressing activity[63]. This leads to the regulation of the ARFs target genes expression. Based on this model, we hypothesised that the impairment of the MMC differentiation in the *spl-1* mutant could be ascribed to the post-translational suppression of ARFs function. Indeed, lower auxin accumulation in *spl-1* may inhibit Aux/IAAs degradation, affecting ARFs activity and, thus, MMC differentiation. Accordingly, ovules containing an Aux/IAA isoform resistant to the auxin-dependent proteasomal degradation might mimic the *spl-1* phenotype. Single-cell RNA-seq, performed using ovules between stages 2-I and 2-III, revealed that Aux/IAA 1, 2, 3, 4, 8, 9, 13, 16 and 26 were expressed in nucellar clusters[64]. To select the appropriate Aux/IAA to repress the ARFs activity in the nucellus, we chose the one that was previously reported to interact with several nucellar-expressed ARFs[65–67], and for which a dominant auxin-resistant allele was available. We chose the Aux/IAA3/SHY2[68] dominant mutant allele *shy2.6*[69]. *shy2.6* carries a mutation in the degron motif, impairing its ability to interact with the SCF/TIR1 complex at threshold auxin levels[69], remaining constitutively bound to ARFs PB1 domain[69]. Consequently, we generated the *pSPL/NZZ::shy2.6* construct to express *shy2.6* in the nucellus of wild-type ovules.

We introduced the *pSPL/NZZ::shy2.6* construct into wild-type plants, obtaining three independent lines expressing the transgene (Supplementary Fig. 11A). Even if with a lower penetrance with respect to the *spl-1* mutant, in all the lines we observed an impairment in MMC differentiation (Fig. 5A–M). To confirm that the observed phenotype was caused by the inhibition of ARFs activity rather than other factors, such as a disruption of the polar auxin transport, we examined the expression of *pPIN1::PIN1:GFP* in *pSPL/NZZ::shy2.6* plants. The expression of PIN1-GFP in *pSPL/NZZ:shy2.6* resembled that of the wild type, suggesting that the observed phenotype was probably caused by an impairment in ARFs activity (Supplementary Fig. 11B–E).

## ARFs activity is required for the MMC differentiation

Collectively, our data support a *scenario* in which the MMC differentiation is mediated by the post-translational regulation of the ARFs action. SHY2 can interact with all the class A ARFs, including ARF5/MP, ARF6, ARF7, ARF8 and ARF19. Concerning the repressor ARFs, SHY2 was shown to interact with ARF4 and ARF9[65–67]. According to an ovule single-cell RNA-seq dataset[64], *MP*, *ARF6*, *ARF8* and *ARF9* are the only SHY2-interacting ARFs expressed in the nucellus. Consequently, we analysed *MP*, *ARF6*, *ARF8* and *ARF9* expression in both wild-type and the *spl-1* ovules. We used previously published reporter lines to study *MP*, *ARF6* and *ARF8* domains of transcription and protein accumulation[70]. Concerning *ARF9*, as no functional protein reporter lines were available, we studied its expression pattern by in-situ hybridisation.

In ovule primordia, the levels of both ARF6-VENUS and ARF8-VENUS were presumably controlled at the post-translational stage, supporting previously reported observations[53] (Supplementary Fig. 12E–L). Indeed, although both genes were transcribed in ovules[64] (Supplementary Fig. 12E, I, G, K), ARF6 was faintly visible solely in a limited number of cells (Supplementary Fig. 12F, H), while ARF8 was not visible (Supplementary Fig. 12J, L). By contrast, the MP-VENUS protein accumulated in the nucellus, as previously shown[70,71] (Supplementary Fig. 12A–D). The three ClassA ARFs exhibited comparable expression and protein accumulation in both wild-type and *spl-1* ovules (Supplementary Fig. 12A–L). Concerning *ARF9*, we detected its transcript accumulation in both wild type and *spl-1* nucellus at stage 1-II (Supplementary Fig. 13A, C). At stage 2-III, the *ARF9* transcript was accumulated in the MMC and the integuments of wild-type ovules (Supplementary Fig. 13B). By contrast, in the *spl-1* mutant background, *ARF9* transcript appeared miss-localised in both the nucellus and the chalaza (Supplementary Fig. 13D). This could be mainly due to the fact that, in a wild-type ovule, *ARF9* transcript accumulates in the MMC and the integuments, both structures that are impaired in the *spl-1* background. Despite this, by analysing the *spl-1* RNA-seq, we observed an overall decrease in *ARF9* expression level in the *spl-1* pistil (Supplementary Data 3).

Given their expression patterns, MP and ARF9 were selected as ARFs potentially involved in the downstream auxin response, required for MMC differentiation. According to our hypothesis, the absence of

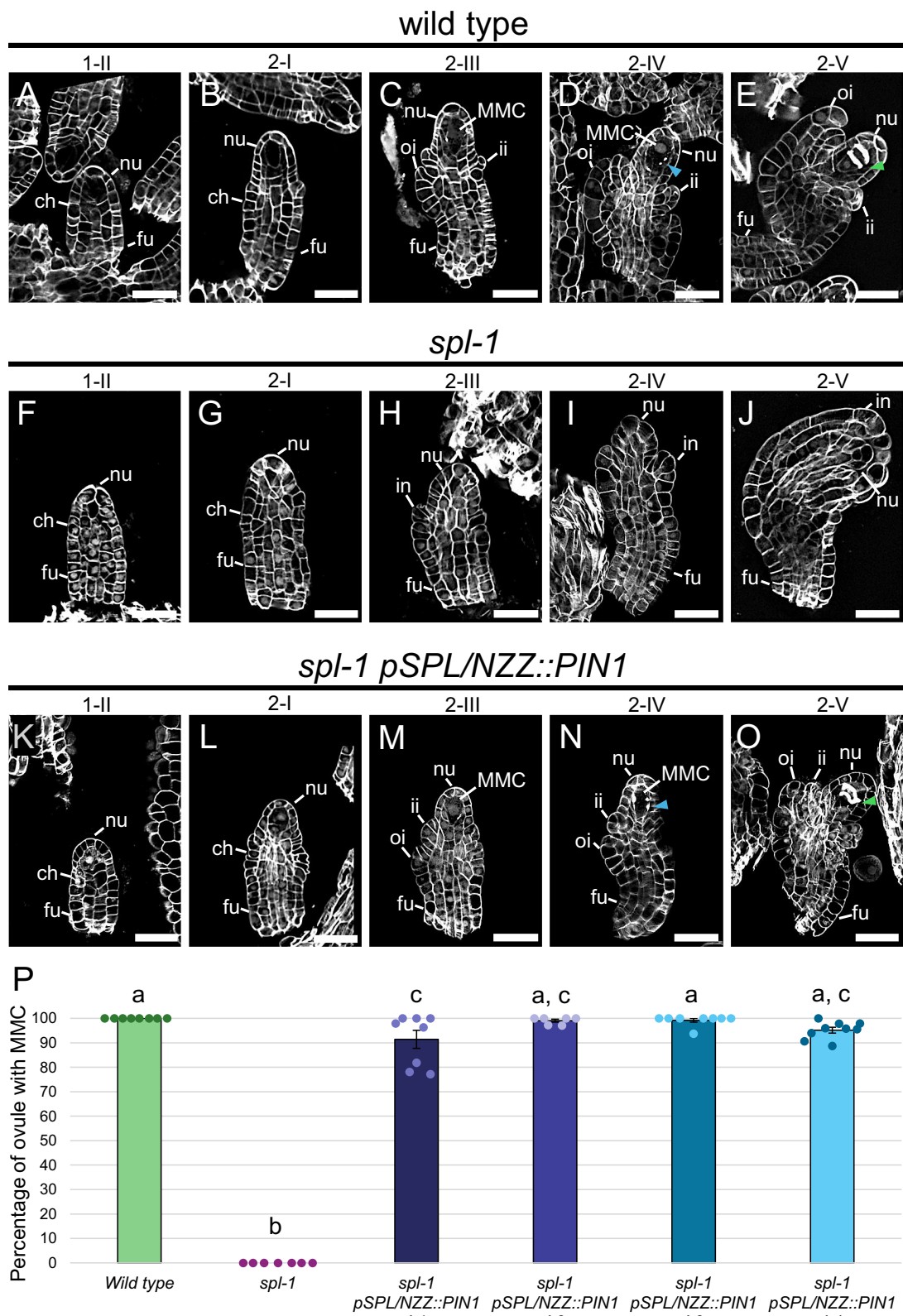

**P**

**wild type** (A–E), **spl-1** (F–J), **spl-1 pSPL/NZZ::PIN1** (K–O), stages 1-II, 2-I, 2-III, 2-IV, 2-V.

the MMC in the *spl-1* mutant was associated with a decrease in the nucellar auxin response, caused by the stabilisation of Aux/IAAs and the consequential ARFs inhibition. Indeed, as previously described[63,70,72,73], ARFs activity is post-translationally regulated by the balance between ARFs and Aux/IAAs relative abundance. For this reason, we hypothesised that increasing *MP* or *ARF9* levels in the *spl-1*

background could overcome the repression operated by the Aux/IAAs, restoring the MMC presence. Therefore, we generated the *pSPL/ NZZ::MP* and the *pSPL/NZZ::ARF9* constructs that we introduced in the *spl-1* mutant. We obtained two independent *spl-1 pSPL/NZZ::MP* lines, while we found three *spl-1 pSPL/NZZ::ARF9* independent lines (Supplementary Fig. 14A, B). In *spl-1 pSPL/NZZ::MP* plants, we observed that

**Fig. 4 | PIN1 expression derived by *SPL/NZZ* regulatory sequences is sufficient to restore the MMC differentiation in the *spl-1* ovule.** Wild type ovules at stage 1-II (**A**), 2-I (**B**), 2-III (**C**), 2-IV (**D**) and 2-V (**E**). At stage 2-III, the MMC is differentiated (**C**) while callose depositions can be observed at stage 2-IV (**D**). At stage 2-V (**E**), the MMC meiosis ends as denoted by the formation of the two meiotic plates. *spl-1* ovules at stage 1-II (**F**), 2-I (**G**), 2-III (**H**), 2-IV (**I**) and 2-V (**J**). In the *spl-1* mutant the MMC do not differentiate (**H, I**). *spl-1 pSPL/NZZ::PIN1* ovules at stage 1-II (**K**), 2-I (**L**), 2-III (**M**), 2-IV (**N**) and 2-V (**O**). Restoring *PIN1* expression in the *spl-1* nucellus is sufficient to drive the MMC differentiation (**M, N**). The MMC can be observed at stage 2-III (**M**). As in the wild-type situation (**D**), the MMC is surrounded by callose depositions at stage 2-IV (**N**) and correctly goes through the meiotic process (**O**). **P** Bar plot showing the mean ± SEM of the percentage of ovules developing the MMC, in wild type, *spl-1* and four *spl-1 pSPL/NZZ::PIN1* independent lines. The analysis has been performed on 8, 7, 8, 6, 8 and 9 different pistils, respectively, for the wild type (202 ovules observed in total), *spl-1* (238 ovules observed in total), *spl-1 pSPL/NZZ::PIN1 L1* (294 ovules observed in total), *spl-1 pSPL/NZZ::PIN1 L2* (196 ovules observed in total), *spl-1 pSPL/NZZ::PIN1 L3* (274 ovules observed in total) and *spl-1 pSPL/NZZ::PIN1 L4* (364 ovules observed in total). Source data are provided as a Source Data file. Letters above the bars indicate homogenous categories with *p* ≤ 0.05 as determined by one-way ANOVA with post-hoc Tukey HSD test. Exact *P* values for each comparison can be found in the Source Data file. Single measures are represented with dots in the bars. nu nucellus, ch chalaza, fu funiculus, ii inner integument, oi outer integument, in integument, MMC megaspore mother cell. Blue arrowheads indicate the callose depositions. Green arrowhead indicates the meiotic plates. Scale bar = 20 μm.

~10% of ovules developed an MMC (Fig. 6A–F, I). However, these cells did not undergo meiosis, suggesting only a partial acquisition of the MMC identity (Supplementary Fig. 14C). By contrast, the majority of *spl-1 pSPL/NZZ::ARF9* ovules differentiated an MMC, which could also proceed into sporogenesis (Fig. 6G, H, J, and Supplementary Fig. 14D), indicating the direct involvement of ARF9 in the MMC differentiation.

Altogether, our results highlight the regulation of the ARFs action as a key step of the SPL/NZZ downstream response during MMC differentiation.

## Discussion

Although the mechanisms necessary to restrict the acquisition of the MMC identity to a single cell have been addressed[74], the regulatory network controlling MMC differentiation is still largely unknown. Since SPL/NZZ is a major factor controlling MMC development, we initially focused on the identification of SPL/NZZ-interacting partners *in planta*. Even though previous works highlighted the TCP TFs as the main SPL/NZZ partners[4,5], in our Co-IP/MS dataset, the MADS-domain TFs resulted in the most abundant TF family. The differences in SPL/NZZ partners identified by the Co-IP/MS and by the previous screenings of SPL/NZZ interactors could be partially explained by our use of the *AP1:GR/ap1cal* inducible system. This system allowed us to find SPL/NZZ partners within a constrained developmental time window. Additionally, while our experiments were performed *in planta*, the previously reported interactors were identified by yeast-two-hybrid screenings[4,5]. We identified only one TCP family member among SPL/NZZ partners. Taking into consideration that a limited number of SPL/NZZ target genes were also identified as TCP targets (Supplementary Data 2), we suggest that complexes including SPL/NZZ and TCPs may function predominantly in other developmental stages or tissues, with respect to the ones analysed in our experiments. On the contrary, several MADS-domain transcription factors were enriched in the SPL/NZZ Co-IP/MS data. Indeed, the genomic sequences bound by SPL/NZZ-containing complexes comprised the CArG-box consensus DNA sequence, typically used by MADS-domain TFs for DNA binding.

We highlighted SEP3 as a TF putatively involved in SPL/NZZ recruitment to the DNA, at least in ovules from stage 2-I to 2-III, since these were the stages from which we had isolated SPL/NZZ-GFP to perform both ChIP-seq and Co-IP/MS. Indeed, SPL/NZZ shared more than 90% of its targets with SEP3, and SPL/NZZ and SEP3 bound these genes at the same genomic positions. To gather additional evidence supporting the cooperation between MADS-domain TFs and SPL/NZZ during MMC differentiation, we developed transgenic lines exhibiting reduced expression of multiple ovule-identity MADS-box genes in the SPL/NZZ expression domain. This strategy was crucial to overcome the redundancy of the ovule identity MADS-domain TFs, and because multiple mutations in these genes result in organ homeotic modifications, hindering the differentiation of ovules. Indeed, higher-order mutants such as *sep1 sep2 sep3* have flower organs converted into sepals[45] and *stk shp1 shp2 ag/+* develop leaves instead of ovules[21,23]. The transgenic plants obtained by post-transcriptionally silencing *SEP* genes in the *shp1 shp2 stk* mutant, using the SPL/NZZ promoter, showed ovules with defects in MMC differentiation. In these transgenic plants, we also observed additional defects resulting from the broader role of these MADS-domain TFs in ovule differentiation and development with respect to SPL/NZZ activity (Supplementary Fig. 5D).

The MADS-domain TFs STK, AG and SEP3 were shown to be fundamental for ovule development[21–23], forming complexes that regulate ovule identity. For instance, the interaction among STK, AG, SEP3 and BEL1 was described as of pivotal importance to determine chalaza identity and to correctly differentiate the integuments[23]. Taking into consideration the present and the previously reported findings, it became evident that MADS-domain TF complexes interact either with SPL/NZZ or BEL1 to control the nucellar and chalazal development, respectively. Indeed, the role of SPL/NZZ in repressing chalaza identity genes in the nucellus, such as *ANT*, *BEL1* and *PHB*, was already reported[12,37]. At the same time, SPL/NZZ ectopic expression was associated with the differentiation of multiple MMC-like cells[11].

We show that the key mechanism for the MMC differentiation likely relies on the post-translational regulation of ARFs activity, associated with the nucellar auxin accumulation.

Re-establishing the auxin transport in the *spl-1* nucellus, by expressing *PIN1* under the SPL/NZZ promoter, was sufficient to restore the MMC differentiation and sporogenesis (Fig. 7D). At the same time, the expression of the auxin-insensitive *shy2.6* mutant protein mimicked the *spl-1* phenotype (Fig. 7E). We propose that SPL/NZZ activity ensures the correct nucellar auxin level and response by indirectly regulating the expression of *PIN1* (Fig. 7A).

Likely, SPL/NZZ, acting together with SEP3 and other MADS-domain TFs, promotes *PIN1* expression by repressing *ANT* and other chalaza genes in the nucellus. According to this, *ANT* ectopic expression in the *spl/nzz* nucellus, together with other chalaza-specific factors, could cause a downregulation of *PIN1* and, consequently, the impaired nucellar auxin accumulation (Fig. 7B). Indeed, it was reported that the *ant* mutation partially rescues the lack of the MMC in the *spl/nzz* mutant[12], suggesting that the nucellar auxin level could also be restored in the *spl/nzz ant* double mutant (Fig. 7C). To define whether *ANT* expression in the nucellus was sufficient to control *PIN1*, we transformed wild-type plants with a *pSPL/NZZ::ANT* construct. Even though we obtained 3 independent lines expressing the construct (Supplementary Fig. 15A), we did not observe defects in MMC development (Supplementary Fig. 15B, C). This result is not unexpected since, in *spl/nzz* mutants, several genes involved in the acquisition of the chalazal identity are ectopically expressed[8,12,49]. Therefore, there are other factors not expressed in the wild-type nucellus, which remain unknown to date, that could work together or in parallel with ANT to impair MMC differentiation when expressed in the nucellus (as in the *spl* mutant). According to our data, the SPL/NZZ-dependent repression of such factors should contribute to the establishment of the nucellar auxin response, which is central for MMC differentiation. The repression of chalazal genes could act on the spatiotemporal expression of *PIN1*. Indeed, besides ANT, BEL1 was also reported to be involved in

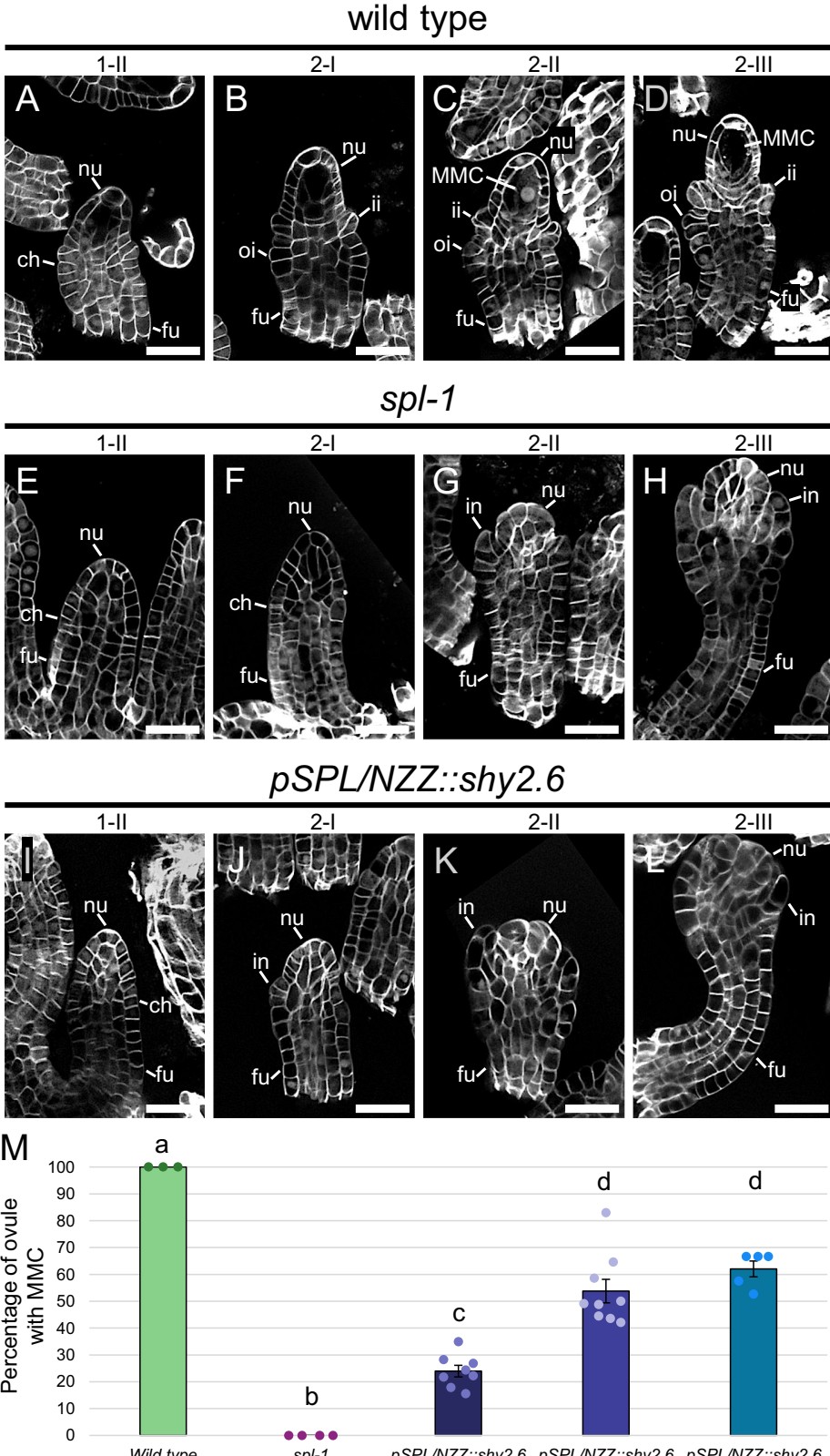

**Fig. 5 | *shy2.6* expression in the nucellus impairs the MMC differentiation.** Wild type ovules at stage 1-II (**A**), 2-I (**B**), 2-II (**C**) and 2-III (**D**). *spl-1* ovules at stage 1-II (**E**), 2-I (**F**), 2-II (**G**) and 2-III (**H**). *pSPL/NZZ:shy2.6* ovules at stage 1-II (**I**), 2-I (**J**), 2-II (**K**) and 2-III (**L**). *shy2.6* expression in the wild-type nucellus impair the MMC differentiation, mimicking the *spl-1* phenotype (**G**, **H**, **K**, **L**). **M** Bar plot showing the mean ± SEM of the percentage of ovules showing an MMC in wild type, *spl-1* and *pSPL/NZZ:shy2.6* lines. The analysis has been performed on 3, 4, 8, 9 and 5 different pistils, respectively, for the wild type (113 ovules observed in total), *spl-1* (157 ovules observed in total), *pSPL/NZZ:shy2.6 L1* (356 ovules observed in total), *pSPL/NZZ:shy2.6 L2* (378 ovules observed in total) and *spl-1 pSPL/NZZ:shy2.6* L3 (202 ovules observed in total). Source data are provided as a Source Data file. Letters above the bars indicate homogenous categories with $p \le 0.05$, as determined by one-way ANOVA with post-hoc Tukey HSD test. Exact *P* values for each comparison can be found in the Source Data file. Single measures are represented with dots in the bars. nu nucellus, ch chalaza, fu funiculus, ii inner integument, oi outer integument, in integument, MMC megaspore mother cell. Scale bars = 20 μm.

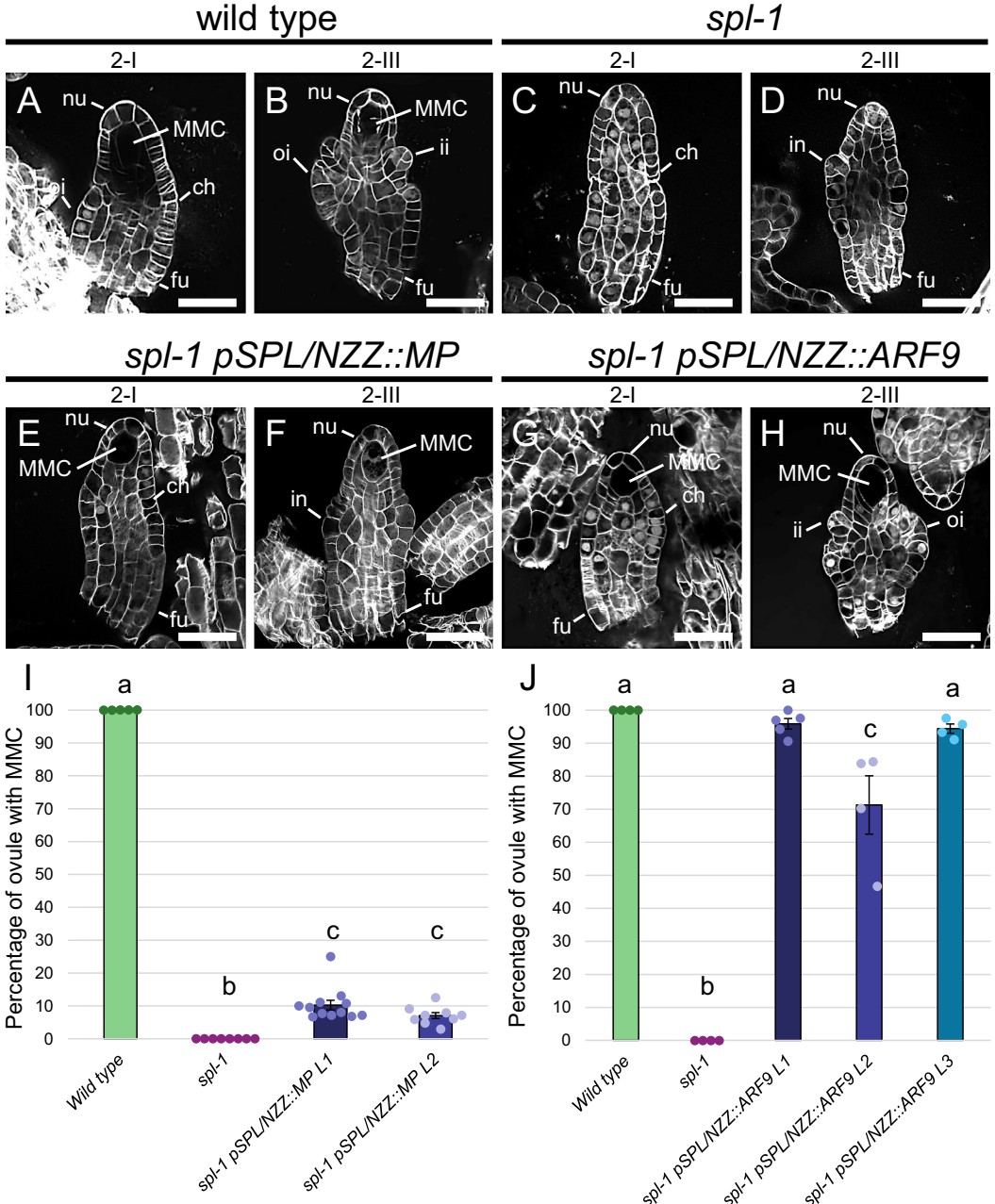

**Fig. 6 | Increasing ARF levels in the *spl-1* nucellus partially restores the MMC differentiation.** Wild type ovules at stage 2-I (**A**) and 2-III (**B**). *spl-1* ovules at stage 2-I (**C**) and 2-III (**D**). *spl-1 pSPL/NZZ::MP* ovules at stage 2-I (**E**) and 2-III (**F**). *spl-1 pSPL/NZZ::ARF9* ovules at stage 2-I (**G**) and 2-III (**H**). While increasing *MP* expression only partially restores the MMC differentiation, *pSPL/NZZ::ARF9* rescues the MMC differentiation in *spl-1*. **I** Bar plot showing the mean ± SEM of the percentage of ovules showing an MMC in wild type, *spl-1* and *spl-1 pSPL/NZZ::MP* lines. The analysis has been performed on 5, 8, 12 and 9 different pistils for the wild type (150 ovules observed in total), *spl-1* (304 ovules observed in total), *spl-1 pSPL/NZZ::MP L1* (362 ovules observed in total) and *spl-1 pSPL/NZZ::MP L2* (270 ovules observed in total). Source data are provided as a Source Data file. Letters above the bars indicate homogenous categories with *p* < 0.05, as determined by one-way ANOVA with post-hoc Tukey HSD test. Exact *P* values for each comparison can be found in the Source

Data file. Single measures are represented with dots in the bars. **J** Bar plot showing the mean ± SEM of the percentage of ovules showing an MMC in wild type, *spl-1* and *spl-1 pSPL/NZZ::ARF9* lines. The analysis has been performed on 4, 4, 5, 4, 4 different pistils for the wild type (157 ovules observed in total), *spl-1* (131 ovules observed in total), *spl-1 pSPL/NZZ::ARF9 L1* (193 ovules observed in total), *spl-1 pSPL/NZZ::ARF9 L2* (115 ovules observed in total) and *spl-1 pSPL/NZZ::ARF9 L3* (177 ovules observed in total). Source data are provided as a Source Data file. Letters above the bars indicate homogenous categories with *p* ≤ 0.05, as determined by one-way ANOVA with post-hoc Tukey HSD test. Exact *P* values for each comparison can be found in the Source Data file. Single measures are represented with dots in the bars. nu nucellus, ch chalaza, fu funiculus, ii inner integument, oi outer integument, in integument, MMC megaspore mother cell. Scale bars = 20 μm.

*PIN1* repression[14]. While ANT regulation over *PIN1* is indirect, no information regarding BEL1 action is available to date.

It is interesting to notice that until stage 2-I, when the MMC starts its differentiation, *PIN1* expression can be observed even in the *spl-1* ovule. In the wild type, starting from stage 2-I until 2-III, the MMC undergoes important modifications. These include a sensible

enlargement in cellular and nuclear dimensions, the deposition of callose, and the expression of cellular-specific markers[48]. These cellular changes are associated with an increase in PIN1-GFP signal in the nucellus L1 and with an increase in auxin signalling output. By contrast, the *spl-1* nucellus, after stage 2-I, showed a significant reduction in both *PIN1* expression and a severely reduced auxin response.

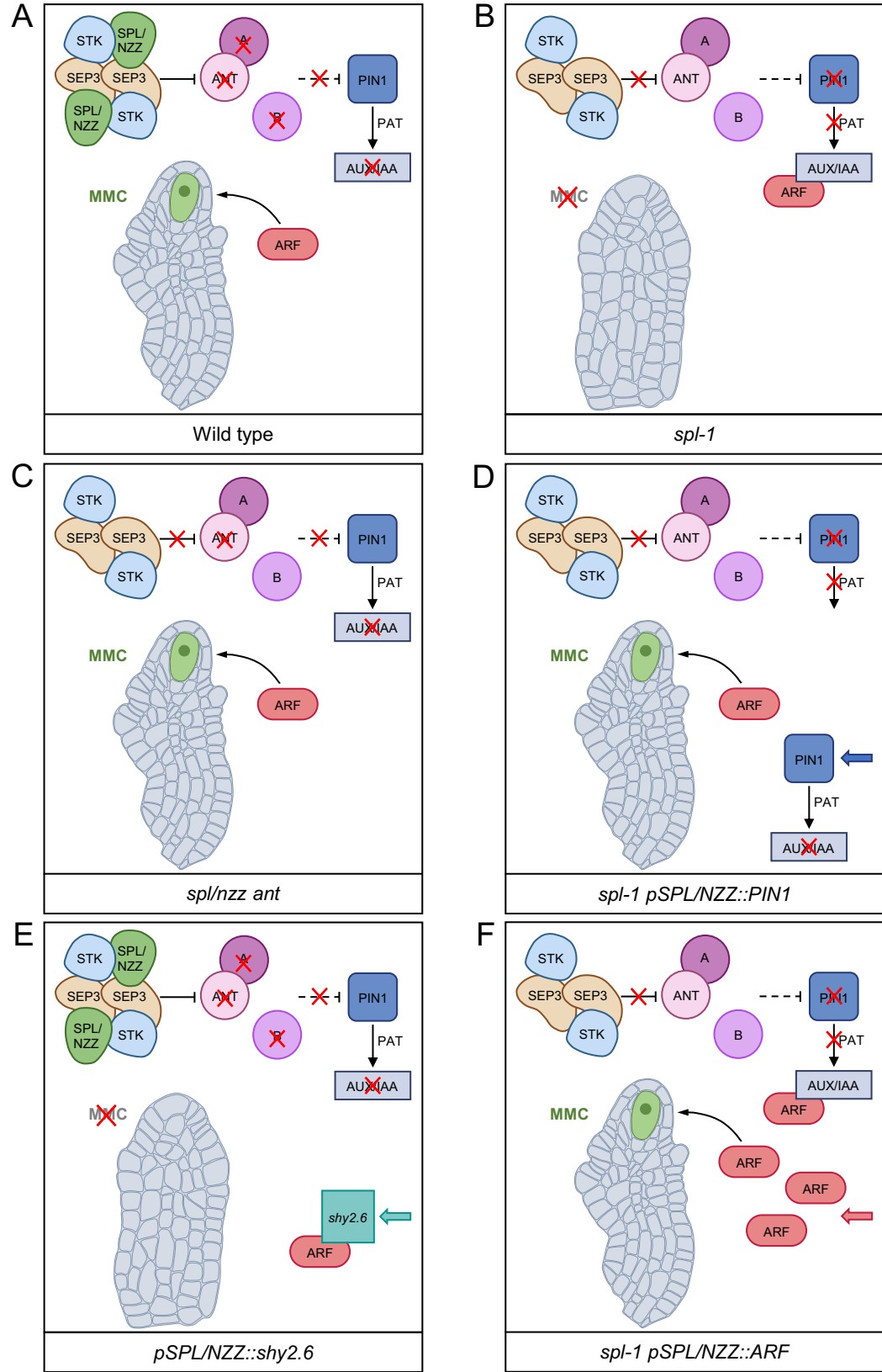

Interestingly, before stage 2-I one or multiple potential MMC initials have been reported to be present[75]. The nucellar phenotypic plasticity is reduced until only one MMC precursor is specified at stage 2-I[75]. The plasticity in the identification of potential MMC candidates is also supported by the eviction of the histone marker HTR13-GFP that, until stage 2-I, occurs in multiple nucellar cells. Removal of HTR13-GFP in one single nucellar cell becomes prominent from stage 2-II, supporting the clear differentiation of the MMC from the other cells at this stage[76].

According to these reported findings, the process of MMC development could be divided into two main steps: the first, during which the nucellar phenotypic plasticity is reduced and leads toward the specification of a single MMC at around stage 2-I. The second step,

**Fig. 7 | SPL/NZZ, by indirectly regulating *PIN1* expression, determine the MMC differentiation through an auxin-dependent network. A** In the wild type, SPL/NZZ interacts with the ovule-identity MADS-domain complex to inhibit *ANT* and other chalazal genes expression, facilitating auxin accumulation in the nucellus and subsequently enabling the ARF-mediated auxin response required to promote the MMC differentiation. **B** In the *spl/nzz* mutant, *ANT* and other chalazal genes are expressed in the nucellus, leading to the downregulation of *PIN1* and to an auxin subthreshold concentration, which disables the post-translational activation of the ARFs, necessary for MMC formation. **C** In the *ant spl/nzz* mutant, the inhibition of

*PIN1* does not take place, allowing the formation of the MMC even in the absence of SPL/NZZ. **D** Expressing *PIN1* under the control of the *SPL/NZZ* promoter in the *spl/nzz* mutant, is sufficient to restore the proper auxin threshold in the nucellus and the ARF-mediated auxin response required for the MMC differentiation. **E** The expression of the non-degradable *shy2.6* isoform in the wild-type nucellus leads to the post-translational deactivation of the ARFs, resulting in the absence of the MMC formation. **F** Increasing ARF levels in the *spl/nzz* mutant can overcome the repressive state caused by Aux/IAAs over-accumulation, determining the MMC differentiation.

during which the MMC differentiates, starts after stage 2-I and ends at stage 2-III. According to our results, the SPL/NZZ-dependent regulation of PIN1 is of pivotal importance for MMC differentiation.

Wild-type and *spl-1* ovules, at precocious developmental stages, present similar phenotypes, allowing the morphological identification of one or multiple potential MMC candidate cells even in the mutant background (Supplementary Fig. 16A–D). This suggests that, although the MMC does not differentiate, putative MMC precursors could be formed in the *spl-1* mutant. This observation is also corroborated by previous published studies[1,2,5], supporting a scenario in which the archespore specification could take place independently from SPL/NZZ action.

The *PIN1* expression during ovule development is involved in several regulatory pathways, specific to each ovule developmental phase, such as ovule primordia formation[52] and integuments formation and outgrowth, and it is likely that SPL/NZZ activity over *PIN1* is temporally restricted to specific processes such as the MMC differentiation.

Previous works already highlighted the connection between the auxin signalling and the MMC initiation. For instance, *ARF3* and *ARF17* silencing were shown to be required to repress the formation of multiple MMC cells[15,77].

According to our findings, auxin accumulation in the nucellus may trigger the degradation of the Aux/IAAs, leading to ARFs activation to support the MMC development (Fig. 7A). Indeed, we observed that inhibiting the ARFs activity in a wild-type situation, by expressing the auxin-insensitive *shy2.6* mutant protein, mimicked the *spl-1* phenotype (Fig. 7E). Among the ARFs expressed in the nucellus, we highlighted MP and ARF9 as putatively involved in the MMC differentiation. We observed that increasing *ARF9* levels could partially rescue the MMC development in *spl-1* (Fig. 7F). According to this, the MMC development might rely on the cooperation between more ARF types. Even though we focused on MP and ARF9, since they both interact with SHY2, we could not exclude the possibility that multiple ARFs could be involved in this process. Indeed, based on ovule single-cell RNA-seq data[64], other repressor *ARFs*, such as *ARF1*, *ARF3* and *ARF18*, are expressed in the nucellus during MMC differentiation, although they do not interact with SHY2. Based on our experiments, we propose that the auxin signalling, and therefore ARFs action, represents the endpoint of SPL/NZZ activity. Despite this, the complete description of how ARFs promote the MMC differentiation is still hindered by the limited knowledge regarding some aspects, such as *ARF9* interacting partners and target genes. This would require additional work that we believe could represent a new and exciting field of study.

In conclusion, our data provide a regulatory mechanism (Fig. 7), in which SPL/NZZ is required for the establishment of the nucellar auxin pattern, needed for the post-translational regulation of ARF TFs responsible for MMC differentiation and development. It will be of great interest to investigate the SPL/NZZ downstream network controlling PMC differentiation. Although we have not investigated this process in detail, we speculate that a similar mechanism could also control PMC formation. Although the SPL/NZZ-interacting MADS-domain partners might be different in anthers with respect to ovules, the *pSPL/NZZ::PIN1* and *pSPL/NZZ::ARF9* constructs also rescued pollen

formation in the *spl-1* mutant, confirming, as previously suggested[9], the importance of auxin response also for the male germline.

## Methods

### Plant material and growing conditions
*A. thaliana* Columbia-0 plants were grown in controlled conditions under a long-day photoperiod (16 h light 150 $\mu m.m^{-2}.s^{-1}$ / 8 h dark) at 22 °C. *spl-1* mutant in the Col-0 background was previously described[14]. The *pSPL/NZZ::SPL/NZZ:GFP* line was previously described[11]. The *AP1:GR/ap1cal* inducible system was described in ref. [17]. *shp1, shp2, stk/STK* plants were described in ref. [21]. *ant.4* mutant in the Ler background was previously described[78,79]. *pSTK:STK:GFP*, *pSEP3::SEP3:GFP*, *pAG::AG:GFP* were previously described[25–27]. The *spl-1 pPIN1::PIN1:GFP* line was described in ref. [14]. *pPIN3::PIN3:GFP*[59], *DR5v2*[58] and *R2D2*[58] lines were manually crossed with *spl-1* heterozygous plants.

### Generation of transgenic lines
To obtain the *pSPL/NZZ::SPL/NZZ:GFP pAP1::AP1:GR ap1cal* inducible line, *pAP1::AP1:GR ap1cal* plants[17] were transformed with the *pSPL/NZZ::SPL/NZZ:GFP* construct via the floral dip technique using the *Agrobacterium tumefaciens* strain GV3101.

*SPL/NZZ:GFP AP1:GR ap1cal* plants were induced with dexamethasone treatment directly on the inflorescence-like meristem for 9 days. After the third and the ninth dexamethasone treatment, the expression of SPL/NZZ-GFP was controlled with a confocal microscope.

The *pSPL/NZZ::MADSas, pSPL/NZZ::PIN1, pSPL/NZZ::shy2.6, pSPL/NZZ::MP, pSPL/NZZ::ARF9* and *pSPL/NZZ::ANT* constructs were obtained with the multisite gateway cloning technique. *MADSas, PIN1, shy2.6, MP, ARF9* and *ANT* CDS sequences were amplified using the Q5 High-Fidelity DNA polymerase (NEB) and cloned into the *pDONR221* plasmid, according to the BP II protocol (Invitrogen, cat. number: 11789020). Multi-site LR reactions were performed using the *pDONR221 P1-P4* containing the SPL/NZZ promoter sequence[11]; the pDONR221 containing either *MADSas, PIN1, shy2.6, MP, ARF9* or *ANT*; the *pDONR221 P3-P2* carrying the *SPL/NZZ 3'UTR* sequence[11] and the destination vector *pH7m34GW*[80], according to the LR II Plus protocol (Invitrogen, cat. number: 12538120). Thanks to the floral dip technique using the *Agrobacterium tumefaciens* strain GV3101, *pSPL/NZZ::PIN1, pSPL/NZZ::MP* and *pSPL/NZZ::ARF9* were transformed into the SPL/NZZ *spl-1* heterozygous background. Four, two and three T2 independent transformant lines were analysed for each transformation, respectively. *pSPL/NZZ::MADSas* was transformed into the *shp1, shp2, stk/STK* background. Three T1 independent transformant lines were analysed. *pSPL/NZZ::shy2.6* and *pSPL/NZZ::ANT* were transformed into wild-type Col-0 plants. Three T1 independent transformant lines were analysed for each transformation. Primers used are listed in Supplementary Data 4.

### RT-qPCR expression analysis
For each sample, 5 inflorescences pooled together (comprising flowers spanning from stage 1 to stage 12) were used for total RNA extraction. RNA extraction was performed using the Macherey-Nagel NucleoSpin RNA plus kit (Item number: 740984.50) according

to the manual instructions. 500 ng of DNA-clean total RNA for each sample was retrotranscribed using the Bio-Rad iScript™ Reverse Transcription Supermix for RT-qPCR (Bio-Rad, cat number: 1708840). cDNA was used for RT-qPCR using the Bio-Rad iTaq Universal SYBR Green Supermix (Bio-Rad, cat number: 1725120). *ACTIN8* was used as a housekeeping normaliser for relative normalised expression calculation. Expression levels were expressed as mean ± SEM among three technical replicates. Primers used are listed in Supplementary Data 4.

### Analysis of ovule morphology by confocal and optical microscopy

Ovule morphology was evaluated by both confocal and optical widefield microscopy. For confocal microscopy, pistils were manually dissected from the flowers and mounted in a solution of 10 µg/ml Renaissance2200 (Renaissance Chemicals, SR2200). Samples were immediately imaged using a Nikon AX confocal microscope. Renaissance2200 was excited at 405 nm, and the emission signal was detected between 440 and 470 nm. For widefield microscopy, inflorescences were collected in a solution of 9:1 ethanol/acetic acid. After 24 h at 4 °C, inflorescences were transferred to a solution of 70% ethanol. Pistils were manually dissected and mounted in chloral hydrate clearing solution before microscopic imaging. Analysis of the percentage of ovules developing the MMC in wild type, *spl-1, pSPL/ NZZ::MADSas, spl-1 pSPL/NZZ::PIN1, pSPL/NZZ::shy2.6, spl-1 pSPL/ NZZ::MP, spl-1 pSPL/NZZ::ARF9* and *pSPL/NZZ::ANT* was performed on ovules at stages 2-II, 2-III.

For fluorescent reporter lines signal imaging, pistils were mounted in the Renaissance2200 solution and imaged using a Nikon AX confocal microscope. mTQ was excited with a 445 nm laser and detected at 480–490 nm. GFP was excited with a 488 nm laser and detected at 520-540 nm. VENUS was excited with a 514 nm laser and detected at 530–560 nm. tdTOMATO was excited with a 514 nm laser and detected at 570-630 nm. Imaging was performed with sequential settings to eliminate signal crosstalk. Confocal images were analysed with the Fiji software[81]. For *PIN1-GFP* and *DR5v2* signal quantification, the Renaissance2200 channel, showing the cellular organisation, has been used to define a region of interest (ROI) comprising the whole L1 layer of the nucellus. The mean GFP and tdTOMATO signals were subsequently measured in the ROI. The *R2D2* signal was analysed as previously described[70]. The tdTOMATO channel has been used to generate a binary mask indicating the nucellar nucelli showing *R2D2* expression. The VENUS and the tdTOMATO mean signals in the single nuclei were subsequently measured.

### Co-IP/MS material generation and data analysis

IP-MS experiments were performed as described before[20]. Essentially, nuclei were isolated from the unfixed plant tissues. Proteins, including native protein complexes, were extracted from nuclear extracts and immunoprecipitated with the GFP antibody using µMACS GFP Isolation Kit (Miltenyi Biotec, cat. number: 130-091-288). Immunoprecipitated proteins were released from the beads with 8 M urea. As control, protein extracts from before the immunoprecipitation were used and processed with the FAST method[82]. Eluted proteins were reduced with DTT, alkylated with IAA and digested with Trypsin/Lys-C Mix, Mass Spec Grade (Promega, cat. number: V5071). Resulting peptides were desalted, dried and resuspended in a loading solution. Peptide concentration was estimated using the NanoDrop One A205 method. Samples were analysed using the EASY-nLC 1200 (Thermo Fisher Scientific™) coupled to the Q Exactive Plus mass spectrometer (Thermo Fisher Scientific™). Peptides were loaded onto the Acclaim PepMap 100 C18 analytical column and separated with a linear solvent B gradient of 5% to 40% over 70 min and 40% to 50% over 5 min. Tandem MS spectra (MS/MS) were acquired in the data-dependent acquisition (DDA) mode using a TOP15 approach.

The RAW peptide sequencing data obtained from the Q Exactive Plus were processed by the MaxQuant (version 1.6.14.0) software. Settings of the MaxQuant were mostly kept default with "label-free quantification (LFQ)", "iBAQ" and "Match between runs" options on. The protein database was *A. thaliana* UniProt UP000006548 without protein isoforms. After the MaxQuant analysis, the output proteinGroups.txt file was taken for further filtering and statistical analysis in the Perseus (version 1.6.15.0) software. The protein LFQ intensity data were $\log_2$-transformed. Proteins identified by only modified peptides ("Only identified by site"), by reverse sequence database ("Reverse") and protein contaminants ("Potential Contaminant") were removed from the analysis. Proteins with LFQ intensity values in at least two biological replicates and proteins with at least two unique peptides were kept. Missing protein LFQ intensity values were replaced only in the input samples by the imputation from the left arm of the LFQ intensity normal distribution (0.3 width, 1.8 down shift, separately for each column) to emulate background protein level. Relative protein amount differences were evaluated by two-sample t-tests with permutation-based FDR correction and S0 parameter set to 1.0. The GO term enrichment analyses were performed with clusterProfiler[83], with default parameters. The statistical significance of enriched categories was determined by the Benjamini−Hochberg (False Discovery Rate-FDR) test[84], considering significant only categories with FDR ≤ 0.05. GO term enrichment analyses were performed on R studio, version 4.4.1.

### ChIP-seq material generation and data analysis

ChIP-seq experiments were performed as described before[34,35]. Specifically, plant tissues were fixed in 1% formaldehyde solution, frozen in liquid nitrogen and ground to a fine powder. Nuclei were isolated through a sequential wash with buffer containing 0.5% Triton X-100 and lysed followed by chromatin sheering by sonication. After chromatin clearing, immunoprecipitation was performed with the GFP antibody (Abcam, ab290) followed by incubation with protein A magnetic beads (Thermo Fisher Scientific™, cat. number: 88845). The immunoprecipitated chromatin was released from the beads and de-crosslinked. The input sample, corresponding to the protein extract from before the immunoprecipitation, was also de-crosslinked and used as a ChIP control. The DNA was purified and concentrated using DNA Clean & Concentrator-5 kit (Zymo Research, cat. number: D4004) following manufacturer's instructions and eluted with 20 µl of the DNA Elution Buffer. The DNA sequencing libraries for both ChIP and Input samples were prepared using ThruPLEX DNA-Seq Kit (Takara Bio, cat. number: R400674) and SMARTer DNA Unique Dual Index Kit (Takara Bio, cat. number: R400697) following manufacturer's instructions. DNA double size selection 100−700 bp was performed using SPRIselect beads (Beckman Coulter, cat. number: B23317). DNA libraries were sequenced on the Illumina sequencer.

Raw data were trimmed using Trimmomatic 0.39[85] with the following parameters: ILLUMINACLIP :1:30:6, LEADING: 3, TRAILING: 3, HEADCROP: 10, AVGQUAL: 20, MINLEN: 35. Mapping was performed using Bowtie 2 (version 2.5)[86] on TAIR10 genome. Duplicated reads were removed using SAMtools (version 1.7)[87]. The peak calling was performed with MACS2 (version 2.2.9.1)[88] with -g 118459858. Peaks enrichment statistical significance was evaluated according to the Benjamini−Hochberg (FDR) test. Peaks were merged using BEDTools (version 2.26.0)[89]. Only peaks present in at least 2 out of 3 biological replicates were considered. Peaks annotation was performed with ChIPseeker[90] with the following parameters: tssRegion comprised between −3000 and 0 and the genomicAnnotationPriority to 5UTR, 3UTR, Exon, Intron, Promoter, Downstream and Intergenic. The GO term enrichment analysis was performed with clusterProfiler[83], with default parameters. The statistical significance of enriched categories was determined by the Benjamini−Hochberg (FDR) test, considering significant only categories with FDR ≤ 0.05. Peak annotation and the

GO term enrichment analysis were performed on R studio, version 4.4.1. The consensus sequence enrichment analysis was performed using the MEME suite[38] with default parameters. The analysis of common peaks between the SPL/NZZ ChIP-seq and the SEP3 ChIP-seq has been performed by using the command subsetByOverlaps on R studio, version 4.4.1.

## In-silico prediction of protein complex structures

Protein complex structure predictions were performed with Alpha-Fold3 algorithm with default parameters. Protein sequences and domains annotation of SPL/NZZ (AT4G27330), STK (AT4G09960), SEP1 (AT5G15800), SEP2 (AT3G02310) and SEP3 (AT1G24260) were retrieved from the Uniprot database. AlphaFold3 models were analysed with the PyMOL software. The complexes confidence-level representations are based on the confidence files provided by Alpha-Fold3 algorithm. The binding-affinity analysis and the visualisation of the SPL/NZZ binding interface with the MADS tetramer was performed with the Prodigy webserver[31–33] using default parameters, at a temperature of 22 °C.

## RNA-seq material generation and data analysis

Three biological replicates of wild-type and *spl-1* pistils at stage 9[47] were collected in the NucleoProtect RNA (Macherey-Nagel, cat. number: 740750.500) and stored at 4 °C for 1 day. RNA extraction was performed using the NucleoSpin RNA kit (Macherey-Nagel).

RNA was sequenced with the Novaseq X, at a depth of 30 M reads with a paired-end sequencing.

Raw data were trimmed using Trimmomatic 0.39[85] with the following parameters: ILLUMINACLIP :1:30:6, LEADING: 3, TRAILING: 3, HEADCROP: 10, AVGQUAL: 20, MINLEN: 35. Mapping was performed with STAR[91], with --alignIntronMax 10000 on TAIR10 genome. Counts were obtained with featureCounts[92] (version 2.0.1), using Araport11 annotation.

Counts were filtered with HTSFilter[93] and the DEGs were obtained using DESeq2[94], considering statistically differentially expressed DEGs with an FDR ≤ 0.05, as determined by the Benjamini–Hochberg (FDR) test. The GO terms enrichment analyses were performed with clusterProfiler[83]. Only categories with an FDR ≤ 0.05, according to the Benjamini–Hochberg (FDR) test were considered statistically significant. DEGs evaluation and GO terms enrichment analyses were performed on R studio, version 4.4.1.

## In-situ hybridisation

In-situ hybridisation was performed as described in refs. 54,95, with some modification. Briefly, developing inflorescences were fixed in Formaldehyde/Ethanol/Acetic Acid (3.7%/50%/5%) solution, dehydrated and embedded in paraffin. Embedded inflorescences were sectioned to 8 μm thickness with a microtome. Dewaxing in histo-clear was followed by rehydration and acidification in 0.2 M HCl for 20 min. Neutralisation occurs in 2X SSC for 10 min, followed by an incubation for 30 min with 1 μg/ml Proteinase K at 37 °C. Glycine (2 mg/ml) was used to incubate sections for 5 min to block Proteinase K, followed by 10 min of post-fixation in 4% formaldehyde. Washing in PBS and dehydrating via an ethanol series occurs before the application of the hybridisation solution (100 μg/ml tRNA; 6X SSC; 3% SDS; 50% formamide, containing -100 ng/μl of antisense DIG-labelled RNA probe). Samples were hybridised overnight at 48 °C and the next day washed in SSC 2X/formamide 50% solution for 2 times for 1 h each. Antibody incubation (Anti-Digoxigenin-AP, Fab fragments, Roche, cat. number: 11093274910) and colour detection (BCIP/NBT Colour Development Substrate, Promega, cat. number: S3771) were performed according to the manufacturer's instructions. *ANT, PIN1* and *ARF9* anti-sense probes were obtained from retrotranscribed RNA extracted from wild-type inflorescences. Primers used are listed in Supplementary Data 4.

## Yeast-two- and three-hybrid assays

Yeast-two- and three-hybrid assays were performed following the protocol from ref. 96 with some modifications.

Both yeast-two- and three-hybrid assays were performed at 28 °C using a single AH109 yeast strain via co-transformation, or AH109 and Y187 yeast strains through co-transformation and mating.

The interactions were tested in selective media lacking leucine, tryptophan, histidine and adenine. AG and STK full-length CDS and SPL/NZZ and SEP3Δ shorter versions (SPL/NZZΔ and SEP3Δ, respectively) were cloned into Gateway vector GAL4 system (pGBKT7 and pGADT7, Clontech; pTFT1[97]). Primers used are described in Supplementary Data 4.

## FRET-FLIM

To test SPL/NZZ, SEP3 and STK interaction by FRET-FLIM, *SPL/NZZΔ*, *SEP3Δ* and *STK* CDS sequences, without the stop codons, were cloned into the *pDONR207* plasmid according to the BP II protocol (Invitrogen, cat. number: 11789020). Thanks to LR II Plus reaction (Invitrogen, cat. number: 12538120), *SEP3Δ* and *STK* CDS were introduced into the *pAB117* (provided by Rüdiger Simon, HHU Düsseldorf), which allow the C-terminus fusion of the *GFP*. *SPL/NZZΔ* was introduced in the *pAB118* (provided by Rüdiger Simon, HHU Düsseldorf), which allows the C-terminus fusion of the *mCHERRY*. *SEP3Δ* was also cloned into the *pAB119* plasmid (provided by Rüdiger Simon, HHU Düsseldorf) which allow the C-terminus fusion with both *GFP* and *mCHERRY*. Plasmids were transformed in the *Agrobacterium tumefaciens* strain GV3101. Tobacco leaf infiltration was performed as previously described[98]. The effect of FRET between GFP and mCHERRY was measured by FLIM of the GFP, using a Nikon ECLIPSE Ni-E A1 confocal equipped with a pulsed laser at 485 nm and taking advantage of the SymPhoTime 64 software (PicoQuant, https://www.picoquant.com/). Each acquisition has been performed reaching a photon count per pixel spanning from 30 to 50. Acquisition of coumarin6 (0.1 mM EtOH 100%), was used as a reference. Coumarin6 has excitation and emission spectra similar to the GFP ones and has a known photon lifetime of around 2.5 ns. The evaluation of its decay was used, through a phasor transformation function, to calculate the Instrument Response. The FLIM results obtained by the SymPhoTime 64 software were corrected with the Instrument Response thanks to the FLIM-Phasor analysis software[99] (FlimLabs, https://www.flimlabs.com), generating a phasor plot in which the different pixels of an image are clustered in relation to their lifetime. This step allows the identification of the nuclei and the evaluation of the GFP photon decay.

## Statistical information

The SPL/NZZ Co-IP/MS experiment has been performed on three independent biological replicates of *pSPL/NZZ::SPL/NZZ:GFP pAP1::AP1:GR ap1cal* inflorescences at 9 DAI. Proteins with LFQ intensity values in 2 out of 3 biological replicates and with at least two unique peptides were kept. Relative protein amount differences were evaluated by two-sample t-tests with permutation-based FDR correction and S0 parameter set to 1.0.

The SPL/NZZ ChIP-seq experiment has been performed on three independent biological replicates of *pSPL/NZZ::SPL/NZZ:GFP pAP1::AP1:GR ap1cal* inflorescences at 9 DAI. Peaks with an FDR ≤ 0.05 and present in at least 2 out of 3 biological replicates were considered. Statistical significance was evaluated according to the Benjamini–Hochberg (FDR) test.

Statistical significance of SPL/NZZ target genes enrichment against AGRIS and PlantGSAD transcriptional factor target lists has been evaluated with the ShinyGo[100] webtool, and only categories with an FDR ≤ 0.05 were considered significant.

The *spl-1* vs wild type RNA-seq experiment has been performed on three independent biological replicates of wild type and *spl-1* pistils at

stage 9 of development. Statistical significance of *spl-1* vs wild-type DEGs was evaluated with the Benjamini–Hochberg (FDR) test and only genes with an FDR ≤ 0.05 were considered significant.

Statistical significance of Gene Ontology enrichment analyses on the results obtained from the SPL/NZZ Co-IP/MS, the SPL/NZZ ChIP-seq and the *spl-1* vs wild type RNA-seq, was evaluated according to the Benjamini–Hochberg (FDR) test. Categories with an FDR ≤ 0.05 were considered significant.

In the FRET-FLIM experiment to test the SEP3-SPL/NZZ interaction, statistical significance of the reduction in the GFP lifetime has been evaluated according to the results of the student's t-test, two-sided distribution, homoscedastic, comparing the GFP lifetime in nuclei expressing SEP3Δ-GFP with the one measured in nuclei expressing SEP3Δ-GFP + SPL/NZZΔ-mCHERRY. The comparison between SEP3Δ-GFP and SEP3Δ-GFP-mCHERRY has been used as a positive control for the experiment. Comparisons with a $p \leq 0.05$ were considered significant. student's t-test calculation has been performed with Microsoft Excell. In the FRET-FLIM experiment to test the STK-SPL/NZZ interaction, statistical significance of the reduction in the GFP lifetime has been evaluated according to the results of the student's t-test, two-sided distribution, homoscedastic, comparing the GFP lifetime in nuclei expressing STK-GFP with the one measured in nuclei expressing STK-GFP + SPL/NZZΔ-mCHERRY. The comparison between SEP3Δ-GFP and SEP3-GFP-mCHERRY has been used as a positive control for the experiment. Comparisons with a $p \leq 0.05$ were considered significant. student's t-test calculation has been performed with Microsoft Excell. For *PIN1-GFP*, *DR5v2*, and *R2D2* signal intensity measurements, statistical significance has been evaluated according to the results of the student's t-test, two-sided distribution, homoscedastic, comparing the mutant with the wild-type. Comparisons with a $p \leq 0.05$ were considered significant. student's t-test calculation has been performed with Microsoft Excell. For the morphological analyses of ovules differentiating an MMC, as well as funiculus length measurements, statistical significance has been evaluated according to the results of one-way ANOVA with post-hoc Tukey HSD test. Comparisons with a $p \leq 0.05$ were considered significant. The analysis has been performed with the webtool: https://astatsa.com/OneWay_Anova_with_TukeyHSD/. Exact *p*-values can be found in the Source Data.

### Reporting summary

Further information on research design is available in the Nature Portfolio Reporting Summary linked to this article.

## Data availability

All data generated in this study are available from the corresponding author upon reasonable request. Source Data underlying Figs. 1–6 and Supplementary Fig. 5, 7, 9, 10, 11, 14, 15 are provided as a Source Data file. Lists of SPL/NZZ interactors and target genes, as well as differentially expressed genes in *spl-1* with respect to the wild type, can be found in the Supplementary Data 1–3. Primers used are listed in Supplementary Data 4.

The mass spectrometry proteomics data have been deposited to the ProteomeXchange Consortium via the PRIDE[101] partner repository with the dataset identifier PXD070409. ChIP-seq raw data have been deposited to the NCBI Sequence Read Archive (SRA) under the accession PRJNA1303074. RNA-seq raw data have been deposited to the NCBI Sequence Read Archive (SRA) under the accession PRJNA1365841. Source data are provided with this paper.

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

## Acknowledgements

AC and LC were supported by MIUR PRIN2017 (20175R447S); CA was supported by MIUR UNIMI PhD fellowship. SM was supported by post-doctoral fellowships from Fundación Alfonso Martín-Escudero (Madrid, Spain) and UNIMI. AM was supported by a predoctoral grant from UNIMI. Research in the Kaufmann lab is funded by DFG grants 512328399, 512328399, 458750707. The Kaufmann lab wishes to thank the Dresden Concept Genome Centre for support with NGS sequencing. Mass Spec was performed using a Q Exactive Plus funded by DFG (428987924). The authors acknowledge support from the University of Milan through the APC initiative. We thank Francesca Lopez for contributing to preliminary plants characterisation. We thank Stefano Buratti for technical support during the FRET-FLIM experiment. We thank Rosanna Petrella and Simona Masiero for the helpful discussion. Part of this work was performed at the Unitech NoLimits, an advanced imaging facility established by the University of Milan. We thank Laura Luoni and Plant Platform, an advanced facility for plant developmental studies established by the University of Milan.

## Author contributions

A.C., C.A., S.M., and L.C. designed the research. A.C., C.A., G.B., C.S., C.M., S.M., A.R., M.S., A.M., V.G., X.X. performed the experiments. C.A., C.S., A.C., G.B., C.M., L.C., and K.K. analysed the data. A.C., C.A., L.C.,

G.B., C.S., and S.M. wrote the manuscript. L.C. and K.K. supported the research.

## Competing interests

The authors declare no competing interests.
