## [Transparent Peer Review file · Nature Communications]

SPOROCTELESS/NOZZLE cooperates with MADS-domain transcription factors to regulate an auxin-dependent network controlling Megaspore-Mother-Cell differentiation

Corresponding Author: Professor Lucia Colombo

Version 0:

Reviewer comments:

Reviewer #1

(Remarks to the Author)

The manuscript by Cavalleri et al. investigates the role of SPL/NZZ in regulating megaspore mother cell (MMC) development by identifying ovule-identity MADS-domain transcription factors as direct targets, co-regulating an auxin-dependent downstream network. SPL/NZZ is recognized as a master regulator in plant reproduction, particularly in the differentiation of MMCs and PMCs. In this manuscript, the authors performed Co-IP/MS of SPL/NZZ and identified several MADS-domain transcription factors as SPL/NZZ-specific interacting proteins during the early MMC stages of ovule development. The authors then verified their interactions in yeast, supporting the formation of multimeric protein complexes. ChIP-seq assays show that SPL/NZZ and the MADS-domain TFs share common targets, over 90%, indicating that they might function in the same pathway to regulate the MMC development. Through Biological Process GO term enrichment analysis done on these target genes, they made a direct link between SPL/NZZ and auxin. Furthermore, using loss-of-function mutants and complementation assays it is shown that the severe developmental defects observed in the MMC differentiation in *spl-1* mutant ovules could be due to the impairments in PIN1-mediated auxin transport and accumulation, and Aux/IAA-ARF module-mediated auxin transcriptional responses. Together the multi-omics, genetic, and cell biological data conclusively demonstrate that SPL/NZZ acting with MADS-domain transcription factors determines the specification and development of the MMC via an auxin-dependent downstream network.

In general, the manuscript reads well and results are presented in appropriate figures. Several aspects warrant further clarification and additional experiments to strengthen the study's claims.

- 1) Line 126: The authors used yeast-two- and three-hybrid assays to confirm the interactions between SPL/NZZ and MADS-domain TFs. To provide more robust evidence of these interactions in their native context, fluorescence resonance energy transfer (FRET) or fluorescence lifetime imaging microscopy (FLIM-FRET) analyses in living plant cells, especially during MMC differentiation, are recommended.
- 2) Line 153: The manuscript references AlphaFold predictions to suggest the formation of SPL/NZZ-SEPs-STK complexes. To substantiate these predictions, detailed docking site features and docking scores should be presented, offering insights into the potential structural configurations of these protein interactions.
- 3) Line 189: With over 90% overlap between SPL/NZZ and SEP3 ChIP-seq peaks, qRT-PCR assays should be conducted on representative target genes, particularly those involved in the ovule specification and development, to validate the ChIP-seq findings.
- 4) Line 219: Why did the authors choose to evaluate auxin transport, accumulation, and response concerning SPL/NZZ activity, but not auxin biosynthesis? The rationale for this is not fully logical, as some data suggest that SPL/NZZ function is closely linked to auxin biosynthesis. RNA-seq assay shows an impairment in auxin biosynthesis in the *spl-1* mutant. SPL/NZZ bound genes include the gene YUCCA6, which might be a direct target of SPL/NZZ, as later data show that SPL/NZZ-ANT module regulation over PIN1 is indirect in ovule development.
- 5) Line 227: The signal of PIN1-GFP in wild-type and *spl-1* ovules should be quantified (Figure 3A-D).
- 6) Line 233: Investigating whether mutants of MADS-domain TFs, such as *sep* mutants, exhibit impaired auxin response similar to the *spl-1* mutant could reveal functional overlaps and distinctions within this regulatory network.
- 7) Line 265: The complementation assays demonstrating PIN1's ability to rescue ovule developmental defects in *spl-1* mutants would benefit from the inclusion of negative controls, such as inactive versions of PIN1, to confirm the specificity of

the observed rescue effects. Additionally, it would be valuable to assess whether the application of exogenous auxin can rescue the developmental defects observed in the *spl-1* mutants. This approach could provide further insights into the role of SPL/NZZ in auxin-mediated pathways.

8) Line 285: Clarifying the rationale for choosing IAA3/SHY2 as a candidate for testing the hypothesis, including its expression pattern during MMC differentiation, would strengthen the study's focus and relevance.

9) Line 292: I do not agree with the sentence "Even if with different percentages, in all the lines we observed an impairment in the MMC differentiation, resembling the *spl-1* phenotype". SHY2 transgenes are able to maintain the MMC formation at a significantly higher ratio than the *spl-1* mutants, as shown in Figure 5M.

10) The authors highlight ARF9 as a critical factor downstream of SPL/NZZ, yet ARF9 expression levels were found to increase in *spl-1* mutants. The authors propose that increasing ARF9 in *spl-1* backgrounds can rescue MMC differentiation, which logically conflicts with the observation of elevated ARF9 expression already present in *spl-1* mutants. This discrepancy suggests a complex post-transcriptional or post-translational regulation not clearly explained, thus weakening the logic of ARF9's functional role in MMC differentiation.

11) PIN1 expression under the SPL/NZZ promoter in *spl-1* mutants rescues MMC differentiation, the mechanism by which PIN1 re-expression bypasses other downstream consequences of SPL/NZZ loss remains unclear. For instance, integument defects were also rescued by PIN1 re-expression, yet funiculus elongation remained impaired. Clearer explanations for these differential rescue effects are required for logical consistency. Assessing whether PIN1 and ARF9 transgenes can restore MMC differentiation defects in MADS-domain TFs mutants, such as *sep* mutants, and whether common target genes shared by SPL/NZZ and MADS-domain TFs are recovered in both PIN1 and ARF9 re-expression, would provide insights into the interplay between these regulatory pathways.

Some textual suggestions:

1) Line 27: mutations in the SPL/NZZ gene lead to...

2) Line 277: AUX/IAAs should be Aux/IAAs, in the whole manuscript.

3) Line 340: a key step of the SPL/NZZ downstream...

4) Line 399: *carf17* mutant should be *arf17* mutant?

5) Line 428: PCM should be PMC?

6) Figure 3M, remove the two dots above the box plot of wild-type data to enhance clarity.

Reviewer #2

(Remarks to the Author)

Reviewer #3

(Remarks to the Author)

The mechanistic basis of female germ line formation in plants remains unclear. Thus, the authors address an important question of interest to scientists working in the fields of plant development and sexual reproduction.

In this paper, the authors investigate the formation of the megaspore mother cell (MMC). SPL/NZZ (SPL) is the earliest known factor controlling this process. In the past, work from several labs has shed some light on the function of SPL, but the genetic and molecular details of the SPL/NZZ downstream response have remained largely elusive. In this paper, the authors present a major advance in our understanding of the underlying mechanism and thus in our knowledge of early germline development.

In a first set of experiments using a variety of omics approaches (Co-IP/MS, ChIP-Seq, RNAseq) in part with the well-established pAP1::AP1:GR ap1cal1 system, the authors identify MADS domain transcription factors as preferred binding partners of SPL. The list of MADS TFs included well-known oocyte regulators such as STK, SEP3 and AG. Y2H assays suggested direct interaction of SPL with the three TFs, and expression studies confirmed the presence of the three TFs in young ovules. ChIP-Seq identified many direct targets of SPL. Interestingly, the authors found a large overlap with binding sites for SEP3. Comparison of RNA-seq data between wild type and *spl-1* revealed that SPL promotes the activity of genes involved in e.g. chromatin remodeling and flower development, while SPL attenuates the expression of genes involved in auxin biosynthesis and signaling.

In a second set of experiments, prompted in part by previously obtained data, the authors focused on investigating the relationship between SPL and auxin signaling. Results from a complex set of genetic and cell biological experiments suggested that SPL mediates MMC formation by repressing ANT in the nucellus, which in turn allows for regular expression levels of PIN1 in the nucellus, accumulation of an auxin maximum at its distal tip, followed by relief of ARF9 repression by degradation of IAA and MMC formation.

I find the results presented in this paper highly interesting and of value for a general audience.

Comments:

1. In my opinion the paper presents two stories that are only partially linked to each other. While the Co-IP/ChIP data are

interesting their biological relevance is not addressed. For example, what is the role of one or several of the TFs interacting with SPL in MMC formation? In addition, the Y2H data suggest direct interaction between the tested TFs. However, Y2H data stem from a heterologous system and can be prone to artefacts. What about some tests (BiFC, FRET) in plant cells?

While the RNA-seq data suggest that SPL is a general repressor of auxin processes, the authors make a link to the second story via the downregulation of PIN1 in *spl-1*. Thus, this connection appears somewhat artificial. The authors find binding of SPL/SEP3 to ANT regulatory sequences. In addition, they find upregulated ANT expression in the *spl-1* dataset. Using these findings to make the link to PIN1 would make a more logical transition to the second part of the paper.

2. There is a concern with the temporal sequence of events when it comes to the downregulation of PIN1 in *spl-1*. The authors report that the PIN1-GFP signal is unaffected until stage 1-II and note that it is reduced by stage 2-III. The presence of an MMC is clearly evident by stage 2-I and its formation has been proposed to occur even earlier (Hernandez-Lagana, et al. 2021 eLife 10, e66031). One would therefore expect PIN1-GFP to be reduced by late stage 1-II. This discrepancy needs to be explained by the authors.

Minor comments:

There are several typos scattered throughout the manuscript and the supplement. Please fix.

Legend to Figure S1. Nucellus is a Latin term. The plural is "nucelli" not "nucella". Please fix.

Reviewer #4

(Remarks to the Author)

The formation of germline cells is fundamental in plants and animals. In Arabidopsis, the SPL/NZZ is required for the formation of both megaspore mother cell (MMC) and pollen mother cell (PMC). However, since its discovery in 1999, the molecular mechanisms by which SPL/NZZ controls MMC differentiation have remained poorly understood. In this study, the authors identified SPL/NZZ interacting proteins and direct target genes using omics approaches. The findings provide compelling evidence from multiple angles that SPL/NZZ specifies MMC fate by regulating auxin signaling, highlighting an important advancement in this field. Overall, the experiments were adequately designed and well executed. The data presented are extensive and beautiful. However, a major concern is the lack of genetic evidence supporting the roles of SEP and ANT genes in MMC specification.

The authors demonstrate that SPL/NZZ forms a regulatory complex together with MADS-box transcription factors, particularly the SEP3. While SEP3, STK, AG, and BEL1 are known to be involved in chalaza and integument development, the specific role of SEP3 in MMC formation remains unclear. The authors acknowledged that gene redundancy and pleiotropic effects make it difficult to assess the necessity of these MADS-box genes for MMC differentiation, but without direct genetic evidence, it needs to be cautious to conclude that "SPOROCTELESS/NOZZLE acts together with MADS-domain transcription factors to regulate an auxin-dependent network controlling the Megaspore Mother Cell development." To circumvent some of the technical challenges, the authors might consider overexpressing SEP3 and conducting a conditional knockout of SEP genes using the SPL/NZZ promoter.

The identification of direct target genes of SPL/NZZ is a significant progress. Still, there is no genetic evidence supporting the function of ANT in MMC formation, despite the observation that ANT loss of function restores MMC specification in the *spl* mutant. It would be valuable to revisit MMC formation in *ant* mutants using molecular markers to understand how this restoration occurs. Is it possible that overexpression of ANT hinders MMC formation?

Additionally, do *spl-1* pSPL/NZZ::PIN1 and *spl-1* pSPL/NZZ::ARF9 plants produce seeds?

"PCM" should be "PMC" in line 428, and "MCC" should be "MMC" in line 548.

Version 1:

Reviewer comments:

Reviewer #1

(Remarks to the Author)

Overall, the authors have addressed most of the previous concerns. However, a few minor issues remain:

1. The FLIM-FRET images in Figure 1 are of low quality, making them difficult to interpret. Please provide higher-resolution images with proper controls.
2. While the AlphaFold prediction offers structural insights, it is computational and not experimentally validated. It would be more appropriate to move this figure to the supplemental section.
3. Please indicate the sample size for each quantification and statistical analysis throughout the manuscript.

Reviewer #2

(Remarks to the Author)

Reviewer #3

(Remarks to the Author)

In this revised version of the manuscript, the authors addressed my main comments in detail.

1. They added FRET-FLIM data from tobacco experiments demonstrating the principal ability of direct interaction between SPL and SEP3 or SPL and STK in a plant cell. A test in Arabidopsis would have been preferable but this may have been technically challenging.

2. In my opinion, the author's response to my concern about the temporal sequence of events regarding the down regulation of PIN1 in *spl-1* has been inadequate. My reasonings are as follows:

- I am confused about how the authors use the terms "identity acquisition/specification" and "differentiation," as these terms refer to two fundamentally different processes. I am also unclear about the exact roles that the authors propose for SPL and PIN1 in these processes. In the abstract, the authors mention that SPL is a principal regulator of MMC formation because its mutant phenotype affects MMC differentiation. Later, they continue to highlight the role of a regulatory network involving SPL/PIN1 in controlling MMC specification, yet they also discuss the role of SPL/PIN1 in MMC differentiation. I suppose the confusion stems from the question of whether the SPL/ANT/PIN1 axis is involved in specifying the MMC or its differentiation following specification, or in both processes. What is the authors's notion? Please clarify.

- At the onset of stage 2-I, a single large MMC is clearly visible in the wild type. Once the single MMC is enlarged further MMC development takes place during subsequent stages. The authors agree that the specification of a single MMC must occur before stage 2-I. The single large MMC is missing in *spl-1*. Based on cell morphology, the argument that MMC-like precursor cells appear during stages 0-III or 1-I in both the wild type and *spl-1* mutant is not convincing (Supplemental Figure 16). The larger cells present in the prospective nucellar region during stages 0/1 may or may not be MMC precursors. They could be cells that are dividing but have not yet undergone cytokinesis. In any case, if SPL critically affects PIN-1 expression during MMC specification, it must also occur before stage 2-I. Otherwise, MMC specification depends on other SPL-regulated factors, and SPL/ANT-regulated PIN1 expression is relevant for MMC differentiation following specification. Do you suggest that a single MMC is specified in *spl-1* but does not enlarge and continue development? Failure in MMC enlargement/further differentiation could be explained by a role for SPL/ANT/PIN1/auxin in growth control (via for example auxin-mediated cell wall changes) and/or polarity control in the MMC. If so the text should clarify that specification of the single MMC is not apparently affected by SPL.

- There are also conflicting reports in the literature. The Colombo lab published data revealing that ectopic expression of SPL result in supernumerary MMCs (Mendes et.al. 2020 Development 147:dev194274). At the same time, other previous papers from the Colombo lab have provided evidence that, upon amiRNA-based downregulation of PIN1 in young ovules, in the *pin1-5* mutant, or in a cytokinin receptor mutant carrying ovules with no detectable PIN1pro:PIN1-GFP expression, early development of the embryo sac is blocked in the vast majority of ovules (Bencivenga et al., 2012, Plant Cell, 24:2886; Ceccato et al., 2013, PLoS ONE, 8:e66148). Thus, these data do not support a role for PIN1 in MMC specification. Additionally, under such conditions, MMC differentiation and meiosis would not be impaired to the extent that meiosis and megaspore formation could not be completed. How can these findings be reconciled with a model in which SPL/ANT/PIN1 control MMC specification?

Minor point:

There are numerous typos scattered throughout the manuscript. Please correct them.

Reviewer #4

(Remarks to the Author)

This resubmission includes a substantial amount of new data and revisions that address the reviewer's main concerns, particularly the genetic evidence supporting the role of SEP genes in MMC formation. The new result shows that the ectopic expression of ANT in the nucellus does not affect MMC formation. The authors propose that ANT functions redundantly with other genes to specify the nucellus identity. An earlier study showed that loss-of-function of ANT only partially restored MMC formation in *spl* mutant, and these MMCs failed to develop into embryo sacs. Thus, it is possible that ANT is not the sole direct target of SPL in inhibiting PIN1 activity. The working models in Figure 7 are somewhat misleading. Furthermore, the authors should discuss how ANT and other genes controlling nucellus identity, together with other target genes contribute to MMC differentiation. Are *pSPL/NZZ::ANT* plants capable of producing functional embryo sacs or achieving normal seed set?

Version 2:

Reviewer comments:

Reviewer #3

(Remarks to the Author)

With the focus on MMC differentiation rather than specification the authors have addressed my main concerns.

Reviewer #4

(Remarks to the Author)

My concerns have been addressed in this resubmission.

Point-by-Point responses to reviewer comments

Reviewer #1 (Remarks to the Author):

The manuscript by Cavalleri et al. investigates the role of SPL/NZZ in regulating megaspore mother cell (MMC) development by identifying ovule-identity MADS-domain transcription factors as direct targets, co-regulating an auxin-dependent downstream network. SPL/NZZ is recognized as a master regulator in plant reproduction, particularly in the differentiation of MMCs and PMCs. In this manuscript, the authors performed Co-IP/MS of SPL/NZZ and identified several MADS-domain transcription factors as SPL/NZZ-specific interacting proteins during the early MMC stages of ovule development. The authors then verified their interactions in yeast, supporting the formation of multimeric protein complexes. ChIP-seq assays show that SPL/NZZ and the MADS-domain TFs share common targets, over 90%, indicating that they might function in the same pathway to regulate the MMC development. Through Biological Process GO term enrichment analysis done on these target genes, they made a direct link between SPL/NZZ and auxin. Furthermore, using loss-of-function mutants and complementation assays it is shown that the severe developmental defects observed in the MMC differentiation in *spl-1* mutant ovules could be due to the impairments in PIN1-mediated auxin transport and accumulation, and Aux/IAA-ARF module-mediated auxin transcriptional responses. Together the multi-omics, genetic, and cell biological data conclusively demonstrate that SPL/NZZ acting with MADS-domain transcription factors determines the specification and development of the MMC via an auxin-dependent downstream network.

In general, the manuscript reads well and results are presented in appropriate figures. Several aspects warrant further clarification and additional experiments to strengthen the study's claims.

1) Line126: The authors used yeast-two- and three-hybrid assays to confirm the interactions between SPL/NZZ and MADS-domain TFs. To provide more robust evidence of these interactions in their native context, fluorescence resonance energy transfer (FRET) or fluorescence lifetime imaging microscopy (FLIM-FRET) analyses in living plant cells, especially during MMC differentiation, are recommended.

We thank the reviewer for the comment. To support the conclusion that SPL not only form a complex with SEP3 and STK, but that it also directly interacts *in planta* with these two TFs

we performed, as suggested by the reviewer, FRET-FLIM using tobacco leaves as expression system. The SPL interacting partners have been identified by a Co-IP/MS experiment, using flower synchronized just before and at MMC differentiation stages. Thanks to the suggestion of the reviewer, we are more confident to conclude that STK and SEP3 not only form a complex with SPL, but they do directly interact during MMC specification. The GFP was used as donor protein, and we fused it to either SEP3 Δ or STK c-terminus. By contrast, the mCHERRY was selected as acceptor and it was fused to SPL Δ c-terminus. FLIM measurements revealed a significant reduction in the GFP lifetime when SEP3 Δ -GFP or STK-GFP were co-expressed with SPL Δ -mCHERRY in tobacco nuclei, confirming SEP3 and STK ability to physically interact with SPL. The results of this experiment are presented in Figure 1, in the Supplementary Figure 3 and in the main text at lines <149-160>.

2) Line 153: The manuscript references AlphaFold predictions to suggest the formation of SPL/NZZ-SEPs-STK complexes. To substantiate these predictions, detailed docking site features and docking scores should be presented, offering insights into the potential structural configurations of these protein interactions.

We agree with the reviewer on the need of additional quantitative data regarding the prediction of the structure of protein complexes formed by SPL, SEPs and STK. Therefore, we performed a binding affinity prediction of the SPL/NZZ-SEPs-STK complexes using the Prodigy webserver. In all cases, we obtained Gibbs free energy and dissociation constant results that validate the reliability of the AlphaFold models. We also obtained a more detailed visualisation of the SPL/NZZ-SEP-STK docking sites, highlighting the residues within the hydrogen bonds range (5.5Å). This information is presented in Figure 1, in Supplementary Figure 4 and in the main text from lines <170-177>

3) Line 189: With over 90% overlap between SPL/NZZ and SEP3 ChIP-seq peaks, qRT-PCR assays should be conducted on representative target genes, particularly those involved in the ovule specification and development, to validate the ChIP-seq findings.

We thank the reviewer for the comment. To further dissect the activity of SPL/NZZ over its direct targets, we analysed the SPL/NZZ-SEP3 common target list in the light of the DEGs obtained from the *spl-1* RNA-seq. We found that 77 SPL/NZZ-SEP3 targets resulted also differentially expressed in the *spl-1* pistil with respect to the wild type (Supplementary Data

3). Nicely, these include genes involved in ovule development such as *ANT*, *LUG*, *AP2*, *CUC3*, *IPT1*, *SEP3*, *AGL15* and *FUL*. Interestingly, all the auxin biosynthetic genes present in the *spl-1* RNA-seq, including also the genes targeted directly by SPL/NZZ, resulted upregulated (Supplementary Data 3).

We described this analysis in the main text at line <252-259>.

4) Line 219: Why did the authors choose to evaluate auxin transport, accumulation, and response concerning SPL/NZZ activity, but not auxin biosynthesis? The rationale for this is not fully logical, as some data suggest that SPL/NZZ function is closely linked to auxin biosynthesis. RNA-seq assay shows an impairment in auxin biosynthesis in the *spl-1* mutant. SPL/NZZ bound genes include the gene *YUCCA6*, which might be a direct target of SPL/NZZ, as later data show that SPL/NZZ-ANT module regulation over *PIN1* is indirect in ovule development.

We agree that we need to add an explanation for the rationale behind the decision to focus on auxin transport, accumulation, and response concerning SPL/NZZ activity, but not auxin biosynthesis. We did not focus on auxin biosynthesis because it was previously described that genes responsible for auxin biosynthesis are expressed in the chalaza region of the ovule, while SPL/NZZ function on MMC specification is played in the nucellus where its protein is localized at that stage. This information is based on the expression pattern of key players in the auxin biosynthetic pathway as *YUCCA 1, 2, 3, 4, 8, 9* and *TAA1* (as reviewed in <https://doi.org/10.1093/jxb/erv256>). Auxin is then conveyed to the ovule tip through its transport in the L1 layer by *PIN1*, where SPL/NZZ localises. Moreover, it was previously described that *PIN1* expression and auxin response was downregulated in the *spl* mutant. We also analysed in detail the expression of auxin biosynthetic genes in the *spl-1* RNA-seq. As described in the main text: <line 258> Interestingly, all the auxin biosynthetic genes present in the *spl-1* RNA-seq, including also the genes targeted directly by SPL/NZZ, resulted upregulated (Supplementary Data 3). Key players of auxin biosynthesis such as *YUCCA 2, 4, 6* and *TAA1*, which are normally expressed in the ovule chalaza, showed enhanced expression. This result is in line with SPL/NZZ proposed activity as a transcriptional repressor, which is also supported by the reduced expression of *YUC2* and *4* in the *spl-D* dominant mutant. After biosynthesis in the chalaza, auxin is polarly transported towards the primordium apex mainly by *PIN1*, generating a region of high signalling output at the nucellus tip. Auxin is then directed toward the L2 layer by *PIN3* and *PIN1* action. Since

it was already reported that *PIN1* was drastically downregulated in *spl-1* ovules, as well as the auxin response, we decided to investigate in more detail auxin transport and response in wild-type and *spl-1* ovules to evaluate their impact on *spl-1* phenotype. <line 269>

We generated new tables, that we added to the Supplemental Data 3, to clearly highlight the expression of auxin biosynthetic genes in *spl-1* with respect to the wild type.

5) Line 227: The signal of PIN1-GFP in wild-type and *spl-1* ovules should be quantified (Figure 3A-D).

We thank the reviewer for the comment. We quantified the PIN1-GFP signal in the L1 layer of the nucellus in both wild type and *spl-1* mutant. As we observed a different PIN1-GFP behaviour at different timepoints between wild type and *spl-1*, we chose to focus on PIN1-GFP intensity at both stage 2-I and 2-III. We added also the analysis of *DR5v2* signal at the same time-points, together with the re-organisation of the corresponding paragraph.

As described in the main text: <line270> Focusing on wild type ovules, we observed that PIN1-GFP signal increased in the nucellus from stage 2-I, the stage in which the differentiation of the MMC initiates, to the stage 2-III, associated with the complete differentiation of the MMC (Figure 3A, B, M). Likewise, we measured a significant increase in the nucellar *DR5v2* signal passing from stage 2-I to stage 2-III (Figure 3E, F, N). This dynamic in *PIN1* and *DR5v2* expression during the MMC differentiation could suggest the requirement of auxin transport and signalling to support the MMC identity acquisition. Indeed, even though we observed that PIN1-GFP accumulated similarly in wild-type and *spl-1* ovules at stage 2-I (Figure 3A, C, M), at stage 2-III, PIN1-GFP signal was drastically reduced in the *spl-1* mutant (Figure 3B, D, M), as previously shown. Likewise, the *DR5v2* signal observed between wild type ovules at stages 2-I and 2-III (Figure 3 E-H, N) was severely reduced in in the *spl-1* background. (Figure 3 E-H, N). Besides PIN1, also PIN3 is involved in auxin transport in the nucellus. Despite this, *pPIN3::PIN3:GFP* signal was localised at the very tip of ovule primordia in *spl-1*, and no differences were observed compared to the wild type (Supplementary Figure 7A-D).

Downregulation of *PIN1* expression and *DR5v2* response suggests that auxin is accumulated at lower levels in *spl-1* nucellus. Indeed, by using the *R2D2* reporter line, we detected a higher accumulation of *DII-VENUS* in the *spl-1* nucellus with respect to the wild type (Figure 3I-L; Supplementary Figure 7E-I).

Considering the behaviour of *PIN1* and *DR5v2*, together with the fact that auxin transport was previously suggested to be required for the MMC specification, we believe that *PIN1*

downregulation could be the main cause of MMC absence in the *spl-1* mutant ovule. <line290>

6) Line 233: Investigating whether mutants of MADS-domain TFs, such as *sep* mutants, exhibit impaired auxin response similar to the *spl-1* mutant could reveal functional overlaps and distinctions within this regulatory network.

We agree with the reviewer regarding the need of additional data to reveal functional overlaps between MADS-domain TFs and SPL/NZZ. <line211> Our data indicate that SPL/NZZ could require the MADS-domain TFs SEP1, 2, and 3, as well as STK and SHPs for MMC specification. This outcome was also predictable, as it has been shown that MADS-domain TFs of classes C, D and E form multimeric complexes to establish ovule tissue and cell identity. Regrettably, these transcription factors exhibit significant redundancy and multiple mutants, as the *shp1 shp2 stk ag/+* one, displays ovules homeotically changed into leaves, whereas the *sep1 sep2 sep3* mutant has all floral organs homeotically changed into sepals. Therefore, to investigate the potential relevance of these MADS-domain TFs in MMC differentiation, we created a *pSPL/NZZ::MADSas* construct, to facilitate post-transcriptional gene silencing of *SEP* genes in the nucellus (Supplementary Figure 5A). We subsequently introduced this construct into *shp1/shp1 shp2/shp2 stk/STK* plants, intending to inhibit the activity of many ovule-identity MADS-domain TFs within the nucellus. We obtained three independent transformant lines expressing the *pSPL/NZZ::MADSas* (Supplementary Figure 5B). Specifically, lines 1 and 2 resulted *shp1 shp2 STK/STK*, whereas line 3 was classified as a triple *shp1 shp2 stk* mutant. Although the phenotype was much less penetrant with respect to that reported in *spl-1*, the transgenic plants *shp1 shp2 stk pSPL/NZZ::MADSas* and *shp1 shp2 STK/STK pSPL/NZZ::MADSas* showed a ratio of 25% to 45% of ovules that failed to produce the MMC (Figure 2E, F) besides an impairment in the organisation of the nucellus (Figure 2E; Supplementary Figure 5C).

Altogether, these data support that SPL/NZZ and MADS-domain TFs cooperate, forming a multimeric complex for the differentiation of the MMC. <line 231>

We think that these results could provide additional support regarding the cooperation between SPL and MADS during the MMC development. The results of this experiment can be found in Figure 2, Supplementary Figure 5 and in the main text from <lines 211-231>.

7) Line 265: The complementation assays demonstrating PIN1's ability to rescue ovule developmental defects in *spl-1* mutants would benefit from the inclusion of negative controls,

such as inactive versions of PIN1, to confirm the specificity of the observed rescue effects. Additionally, it would be valuable to assess whether the application of exogenous auxin can rescue the developmental defects observed in the *spl-1* mutants. This approach could provide further insights into the role of SPL/NZZ in auxin-mediated pathways.

The *spl-1* complementation by *PIN1* expression is supported by multiple evidence. In first instance, we analysed different independent lines, with the aim to avoid that the final outcome of the experiment could be altered by possible biases, as the position of insertion of the transgene in the genome. Secondly, we used the same *SPL/NZZ* regulatory sequences and expression system to drive the expression of other cDNAs sequences such as *MP* one, without obtaining a consistent rescue of *spl* phenotype. Therefore, we are confident that the *SPL/NZZ* regulatory sequences in the *spl-1* background does not, by itself, complement the *spl* phenotype.

Concerning the application of exogenous auxin to rescue the MMC development in the *spl-1* mutant, we have done the experiments obtaining negative results. Inflorescences were treated with 100µM IAA and collected for DIC microscopy after 24h from the treatment. Despite this, no differences were observed between mock and treated *spl-1* ovules. We don't believe that these results will add useful information to the manuscript since, we think that the problem in the *spl* mutant is the incapability to reach a threshold in auxin concentration in specific nucellar cells due to the severe downregulation of PIN1.

8) Line 285: Clarifying the rationale for choosing IAA3/SHY2 as a candidate for testing the hypothesis, including its expression pattern during MMC differentiation, would strengthen the study's focus and relevance.

We thank the reviewer for the comment, and we are sorry for the lack of clarity. The main objective of the experiment was to impair the MMC differentiation by repressing the ARFs activity. To do so we thought to took advantage of the constitutive inhibitory features of an Aux/IAA dominant mutant variant. We described in more detail the rational of our choice in the main text as follow: <line352> Single-cell RNA-seq, performed using ovules between stages 2-I and 2-III, revealed that Aux/IAAs 1, 2, 3, 4, 8, 9, 13, 16 and 26 were expressed in nucellar clusters. To select the appropriate Aux/IAA to repress the ARFs activity in the nucellus, we chose the one that was previously reported to interact with several ARFs expressed in the nucellus, and for which was available a dominant auxin-resistant allele. We choose the Aux/IAA3/SHY2 dominant mutant allele, named *shy2.6*. *shy2.6* carries a mutation in the degron motif,

impairing its ability to interact with the SCF/TIR1 complex at threshold auxin level, remaining constitutively bound to ARFs PB1 domain. Consequently, we generate *the pSPL/NZZ::shy2.6* construct to express *shy2.6* in the nucellus of wild-type ovules. <line361>

9) Line 292: I do not agree with the sentence “Even if with different percentages, in all the lines we observed an impairment in the MMC differentiation, resembling the *spl-1* phenotype”. *SHY2* transgenes are able to maintain the MMC formation at a significantly higher ratio than the *spl-1* mutants, as shown in Figure 5M.

We are sorry for the misleading phrasing and thank the reviewer for the observation. We modified the main text as follow: <line363> Even if with a lower penetrance with respect to the *spl-1* mutant, in all the lines we observed an impairment in the MMC differentiation, resembling the *spl-1* phenotype (Figure 5A-M). <line364>

10) The authors highlight *ARF9* as a critical factor downstream of *SPL/NZZ*, yet *ARF9* expression levels were found to increase in *spl-1* mutants. The authors propose that increasing *ARF9* in *spl-1* backgrounds can rescue MMC differentiation, which logically conflicts with the observation of elevated *ARF9* expression already present in *spl-1* mutants. This discrepancy suggests a complex post-transcriptional or post-translational regulation not clearly explained, thus weakening the logic of *ARF9*'s functional role in MMC differentiation.

We thank the reviewer for the comment. We totally agree that the description related to *ARF9* expression has to be rephrased. Indeed, *ARF9* is expressed in ovule, as shown by the ISH, in both wild type and *spl*. Despite this, in the *spl-1* RNA-seq, we observed a decrease in *ARF9* expression level in the *spl-1* pistil. This finding corroborates, as reviewer 1 suggested, that the activity of *ARF9* may be regulated by complex post-translational regulations. Indeed, if *ARF9* activity depends on an Aux/IAAs-dependent post-translational regulation, increase the amount of *ARF9* protein in the nucellus at the right stages is supposed to rescue, at least partially, *spl* mutant phenotype, as we have observed. We agree with the reviewer1 that this concept has to be better explained and commented. We modified the main text to increase the clarity of this point <line 391-399>.

11) *PIN1* expression under the *SPL/NZZ* promoter in *spl-1* mutants rescues MMC differentiation, the mechanism by which *PIN1* re-expression bypasses other downstream consequences of *SPL/NZZ* loss remains unclear. For instance, integument defects were

also rescued by PIN1 re-expression, yet funiculus elongation remained impaired. Clearer explanations for these differential rescue effects are required for logical consistency.

Thank you for the suggestion. We have modified the text and added additional experiments to better explain the different rescue defects.

Regarding the rescue of integuments development in *spl-1 pSPL/NZZ::PIN1* ovules we discussed as follow: <line327> ovules could correctly differentiate inner and outer integuments (Figure 4G-J, L-O). Complementation of inner integuments development suggests that SPL/NZZ is important for the proper PIN1 localisation and auxin transport for inner integuments. Indeed, regulation of auxin transport was shown to be involved in integuments formation and differentiation <line330>.

Concerning the funiculus length, <line332> We hypothesised that this phenotype could be related to the fact that genes involved in funiculus differentiation remained deregulated in *spl-1 pSPL/NZZ::PIN1* ovules. For instance, it was shown that *ANT* ectopic expression is directly associated with the differentiation of a longer funiculus. By ISH, we observed that *ANT* remained ectopically expressed in both nucellus and funiculus of *spl-1 pSPL/NZZ::PIN1* ovules (Supplementary Figure 10C). The ectopic expression of *ANT* in *spl-1 pSPL/NZZ::PIN1* not only provide a hierarchical view in which *PIN1* is placed downstream *ANT* but also represent a plausible explanation for the elongated funiculus of *spl-1 pSPL/NZZ::PIN1* ovules. <line340>

Assessing whether PIN1 and ARF9 transgenes can restore MMC differentiation defects in MADS-domain TFs mutants, such as *sep* mutants, and whether common target genes shared by SPL/NZZ and MADS-domain TFs are recovered in both PIN1 and ARF9 re-expression, would provide insights into the interplay between these regulatory pathways.

We agreed with reviewer 1 that the use of *PIN1* and *ARF9* constructs to rescue the MMC differentiation in *sep* mutants would provide insights into the interplay between these regulatory pathways. However, it is important to consider that the multiple *sep* mutants develop only sepals instead of flower organs while the complete loss of ovule identity in the *stk shp2, shp1 ag/+* background determines the conversion of ovules into leaves. Thanks to the *pSPL/NZZ::MADSas* lines, we provided evidence that directly links the MADS function to the MMC formation. Despite this, we could only achieve a partial loss of MMC differentiation, making the analysis of the potential effect of *PIN1* and *ARF9* on MMC development in these lines difficult to evaluate. We discussed the fact that SPL action is,

probably, mediated by multiple redundant MADS <line 441-457>. Likewise, we discussed that understanding ARF9, and ARFs in general, mode of action and interplay with the other components of the MMC developmental pathway will require additional work that will represent a completely new research topic <line 545-550>.

Some textual suggestions:

1) Line 27: mutations in the SPL/NZZ gene lead to...

We modified as suggested

2) Line 277: AUX/IAAs should be Aux/IAAs, in the whole manuscript.

We modified as suggested

3) Line 340: a key step of the SPL/NZZ downstream...

We modified as suggested

4) Line 399: *carf17* mutant should be *arf17* mutant?

It should be indeed *carf17*, as described in the manuscript in which this mutant is presented

5) Line 428: PCM should be PMC?

We modified as suggested

6) Figure 3M, remove the two dots above the box plot of wild-type data to enhance clarity.

We modified the Figure 3 introducing a new line graph to show PIN1-GFP intensity. The previous 3M graph is now placed in the Supplementary Figure 7 and we removed the outlier points from the graph as suggested

Reviewer #2 (Remarks to the Author):

We thank the reviewer for its evaluation of our manuscript

Reviewer #3 (Remarks to the Author):

The mechanistic basis of female germ line formation in plants remains unclear. Thus, the authors address an important question of interest to scientists working in the fields of plant development and sexual reproduction.

In this paper, the authors investigate the formation of the megaspore mother cell (MMC). SPL/NZZ (SPL) is the earliest known factor controlling this process. In the past, work from several labs has shed some light on the function of SPL, but the genetic and molecular details of the SPL/NZZ downstream response have remained largely elusive. In this paper, the authors present a major advance in our understanding of the underlying mechanism and thus in our knowledge of early germline development.

In a first set of experiments using a variety of omics approaches (Co-IP/MS, ChIP-Seq, RNAseq) in part with the well-established pAP1::AP1:GR ap1cal1 system, the authors identify MADS domain transcription factors as preferred binding partners of SPL. The list of MADS TFs included well-known oocyte regulators such as STK, SEP3 and AG. Y2H assays suggested direct interaction of SPL with the three TFs, and expression studies confirmed the presence of the three TFs in young ovules. ChIP-Seq identified many direct targets of SPL. Interestingly, the authors found a large overlap with binding sites for SEP3. Comparison of RNA-seq data between wild type and spl-1 revealed that SPL promotes the activity of genes involved in e.g. chromatin remodeling and flower development, while SPL attenuates the expression of genes involved in auxin biosynthesis and signaling.

In a second set of experiments, prompted in part by previously obtained data, the authors focused on investigating the relationship between SPL and auxin signaling. Results from a complex set of genetic and cell biological experiments suggested that SPL mediates MMC formation by repressing ANT in the nucellus, which in turn allows for regular expression levels of PIN1 in the nucellus, accumulation of an auxin maximum at its distal tip, followed by relief of ARF9 repression by degradation of IAAs and MMC formation.

I find the results presented in this paper highly interesting and of value for a general

audience.

Comments:

1. In my opinion the paper presents two stories that are only partially linked to each other. While the Co-IP/ChIP data are interesting their biological relevance is not addressed. For example, what is the role of one or several of the TFs interacting with SPL in MMC formation? In addition, the Y2H data suggest direct interaction between the tested TFs. However, Y2H data stem from a heterologous system and can be prone to artefacts. What about some tests (BiFC, FRET) in plant cells?

We thank the reviewer for comment and suggestions. Even if the centrality of SPL/NZZ action during MMC development was well known since its first description, so far, its molecular mode of action was still poorly understood. One of the greatest advances in this field has been proposed by Wei and colleagues (2015), when they demonstrated that SPL do not have a DNA binding domain and works as a corepressor of the transcription by interacting with TFs.

With our work we confirm that SPL binding to the DNA depends on its recruitment into TF complexes. Indeed, we have identified that a large portion of SPL direct targets in ovules at around stage 2-I to stage 2-III, which are the stages from which we isolated SPL/NZZ-GFP to perform ChIP-seq (using synchronised inflorescences at 9DAI), are bound at the level of CarG box sequences, as a consequence of SPL interaction with MADS-domain TF complexes, at least at this stage of development.

Even if many of SPL-interacting partners as SEP3, STK, SHPs and AG, were shown to be fundamental for ovule formation, their involvement in the MMC definition was not known. Moreover, the discovery of the involvement of these TFs in the germline specification could open up new line of research, as their study during the male germline formation.

Concerning the generation of additional data to confirm the interaction between SPL and factors as SEP3 and STK *in-planta*, we performed FRET-FLIM using tobacco leaves as expression system as suggested also by reviewer 1. The GFP was used as donor protein, and we fused it to either SEP3 Δ or STK c-terminus. By contrast, the mCHERRY was selected as acceptor and it was fused to SPL Δ c-terminus. FLIM measurements revealed a significant reduction in the GFP lifetime when SEP3 Δ -GFP or STK-GFP were co-expressed

with SPLΔ-mCHERRY in tobacco nuclei, confirming SEP3, STK ability to physically interact with SPL. The results of this experiment are presented in Figure1, in the Supplementary Figure 3 and in the main texts at lines <149-160>

While the RNA-seq data suggest that SPL is a general repressor of auxin processes, the authors make a link to the second story via the downregulation of PIN1 in *spl-1*. Thus, this connection appears somewhat artificial. The authors find binding of SPL/SEP3 to ANT regulatory sequences. In addition, they find upregulated ANT expression in the *spl-1* dataset. Using these findings to make the link to PIN1 would make a more logical transition to the second part of the paper.

We thank the reviewer, and we are sorry if the rationale behind the link between SPL and the auxin signalling has appeared uncertain. To understand the biological pathway associated with SPL/NZZ activity our first approach has been based on the analysis of the GO categories enriched in our omics datasets. Ontology analysis pointed out the connection between SPL/NZZ and the auxin signalling. This association has not been unexpected, as previous works already pointed out the requirement of auxin transport and response for the MMC differentiation. Likewise, the downregulation of *PIN1* in the *spl* mutant background was already described. After having clarified whether *PIN1* downregulation could have been a crucial factor associated with the absence of the MMC in the *spl* ovule, our logical approach has been to investigate a possible molecular mechanisms that could link SPL/NZZ to *PIN1*. The absence of *PIN1* among SPL/NZZ direct targets, together with its downregulation in the mutant ovule, clearly indicated the existence of one or more intermediate that connect SPL/NZZ to *PIN1*. Starting from these finding, we thought to searched, among SPL direct targets, factors which could be genetically associated with *SPL*, and which could work as intermediary of SPL/NZZ action over *PIN1*. Our attention was caught by *ANT*, especially due to the existence of previous works reporting the ability of *ant* mutation to partially rescue the MMC differentiation in the *spl* background.

Despite this, no data connecting ANT activity to *PIN1* expression were known, hindering the directly link between *ANT* ectopic expression to *PIN1* miss-regulation in *spl-1*. Indeed, to make this connection has required us testing whether *ANT* could repress *PIN1*. By ISH, we show that ANT could indeed work as repressor of *PIN1*, as indicated by the expansion in *PIN1* expression domain in the *ant-4* mutant. We modified the main text <line 294-314> to enhance the clarity.

2. There is a concern with the temporal sequence of events when it comes to the downregulation of PIN1 in *spl-1*. The authors report that the PIN1-GFP signal is unaffected until stage 1-II and note that it is reduced by stage 2-III. The presence of an MMC is clearly evident by stage 2-I and its formation has been proposed to occur even earlier (Hernandez-Lagana, et al. 2021 eLife 10, e66031). One would therefore expect PIN1-GFP to be reduced by late stage 1-II. This discrepancy needs to be explained by the authors.

We thank the reviewer for rising this very interesting point. As the reviewer suggested, the MMC starts its differentiation from stage 2-I. Despite this, the MMC differentiation is a dynamic process that span from stage 2-I to stage 2-III. Indeed, in this time frame, the MMC undergoes to important modifications. These include sensible enlargement in cellular and nuclear dimensions, the deposition of callose, and the expression of cellular specific markers. Indeed, some features as callose punctuation are visible only from stage 2-II and especially at stage 2-III. According to this, the MMC differentiation has to be considered a dynamic process that initiates and conclude across different ovule developmental stages. As shown by Hernandez-Lagana, et al. 2021, before the start of the MMC differentiation, in the nucellus one or multiple potential MMC initial are present. Over time, the nucellar phenotypic plasticity is reduced until only one MMC precursor is present, at stage 2-I. The plasticity in the identification of potential MMC candidates is also supported by the eviction of the histone marker HTR13-GFP that, until stage 2-I, occurs in multiple nucellar cells. Eviction of HTR13-GFP in one single nucellar cell become prominent from stage 2-II, supporting the clear differentiation of the MMC from the other cells. Despite this, it is interesting to notice that wild type and *spl-1* ovules, at precocious developmental stages, present similar phenotypes allowing the morphological identification of one or multiple potential MMC candidate cells even in the mutant background (Supplementary Figure 16A-D). This suggest that, although the MMC do not differentiate, the formation of putative MMC precursors is not affected in the *spl-1* mutat (Supplementary Figure 16A-D). According to this, the process of MMC development could be divided into two main steps: the first one would be the formation of one or multiple MMC candidates, an event which could happen independently from SPL/NZZ action. This step is limited by the reduction of the nucellar phenotypic plasticity toward the selection of one single MMC candidate that occur around stage 2-I. The second step would be the SPL/NZZ-dependent MMC identity acquisition, which starts from stage 2-I until MMC full differentiation and separation from the other

nucellar cells at stage 2-III. According to this, SPL/NZZ action could be temporarily restricted during the MMC identity acquisition phase, which correspond to the step in which we observed the major changes in *PIN1* expression between wild type and *spl-1* ovules.

<line519>

We discussed this point in the discussion section <line 491-525>, and we introduced an additional Supplementary Figure (Supplementary Figure 16) to show wild type and *spl-1* ovules phenotype ad precocious developmental stages.

Minor comments:

There are several typos scattered throughout the manuscript and the supplement. Please fix.

Thank you we fixed the typos

Legend to Figure S1. Nucellus is a Latin term. The plural is “nucelli” not “nucella”. Please fix.

Thank you, we fixed the error

Reviewer #4 (Remarks to the Author):

The formation of germline cells is fundamental in plants and animals. In Arabidopsis, the SPL/NZZ is required for the formation of both megaspore mother cell (MMC) and pollen mother cell (PMC). However, since its discovery in 1999, the molecular mechanisms by which SPL/NZZ controls MMC differentiation have remained poorly understood. In this study, the authors identified SPL/NZZ interacting proteins and direct target genes using omics approaches. The findings provide compelling evidence from multiple angles that SPL/NZZ specifies MMC fate by regulating auxin signaling, highlighting an important advancement in this field. Overall, the experiments were adequately designed and well executed. The data presented are extensive and beautiful. However, a major concern is the lack of genetic evidence supporting the roles of SEP and ANT genes in MMC specification.

The authors demonstrate that SPL/NZZ forms a regulatory complex together with MADS-

box transcription factors, particularly the SEP3. While SEP3, STK, AG, and BEL1 are known to be involved in chalaza and integument development, the specific role of SEP3 in MMC formation remains unclear. The authors acknowledged that gene redundancy and pleiotropic effects make it difficult to assess the necessity of these MADS-box genes for MMC differentiation, but without direct genetic evidence, it needs to be cautious to conclude that “SPOROCTELESS/NOZZLE acts together with MADS-domain transcription factors to regulate an auxin-dependent network controlling the Megaspore Mother Cell development.” To circumvent some of the technical challenges, the authors might consider overexpressing SEP3 and conducting a conditional knockout of SEP genes using the SPL/NZZ promoter.

We thank the reviewer, and we agree with its evaluation. <line210> Our data indicate that SPL/NZZ could require the MADS-domain TFs SEP1, 2, and 3, as well as STK and SHPs for MMC specification. This outcome was also predictable, as it has been shown that MADS-domain TFs of classes C, D and E form multimeric complexes to establish ovule tissue and cell identity. Regrettably, these transcription factors exhibit significant redundancy and multiple mutants, as the *shp1 shp2 stk ag/+* one, displays ovules homeotically changed into leaves, whereas the *sep1 sep2 sep3* mutant has all floral organs homeotically changed into sepals. Therefore, to investigate the potential relevance of these MADS-domain TFs in MMC differentiation, we created a *pSPL/NZZ::MADSas* construct, to facilitate post-transcriptional gene silencing of *SEP* genes in the nucellus (Supplementary Figure 5A). We subsequently introduced this construct into *shp1/shp1 shp2/shp2 stk/STK* plants, intending to inhibit the activity of many ovule-identity MADS-domain TFs within the nucellus. We obtained three independent transformant lines expressing the *pSPL/NZZ::MADSas* (Supplementary Figure 5B). Specifically, lines 1 and 2 resulted *shp1 shp2 STK/STK*, whereas line 3 was classified as a triple *shp1 shp2 stk* mutant. Although the phenotype was much less penetrant with respect to that reported in *spl-1*, the transgenic plants *shp1 shp2 stk pSPL/NZZ::MADSas* and *shp1 shp2 STK/STK pSPL/NZZ::MADSas* showed a ratio of 25% to 45% of ovules that failed to produce the MMC (Figure 2E, F) besides an impairment in the organisation of the nucellus (Figure 2E; Supplementary Figure 5C).

Altogether, these data support that SPL/NZZ and MADS-domain TFs cooperate, forming a multimeric complex for the differentiation of the MMC. <line231>

The results of this experiment can be shown in Figure 2, Supplementary Figure 5 and in the main text from <lines 211-231>.

The identification of direct target genes of SPL/NZZ is a significant progress. Still, there is no genetic evidence supporting the function of ANT in MMC formation, despite the observation that ANT loss of function restores MMC specification in the *spl* mutant. It would be valuable to revisit MMC formation in *ant* mutants using molecular markers to understand how this restoration occurs. Is it possible that overexpression of ANT hinders MMC formation?

We observed that the MMC is specified in the *ant* mutant background and correctly divide through meiosis as was also previously reported (Losa et al., 2010). However, we agree with the reviewer on the need of additional studies to clarify precisely how ANT regulates late sporogenesis and gametogenesis. The main *ant* defects reported are the lack of gametogenesis and the impaired integuments initiation probably due to a miss regulation of *PIN1* and hormonal homeostasis in the chalaza. This is a current line of our research. As point out by the reviewer, the genetic association between *ANT* and *SPL* during the MMC development was already known since *ant* mutation could rescue partially the *spl* mutation. In addition, we were able to highlight *ANT* ability to suppress the expression of *PIN1*. The role of ANT in regulating *PIN1* in ovules can be support by the data included in our article and the analysis of *ant* mutants that, likely due to ectopic expression of *PIN1*, fails to proper accumulate auxin to form the integuments (as lateral structure of the chalaza)

Nevertheless, to gather additional data on *ANT* relevance for the MMC specification, we followed the reviewer suggestion regarding its ectopic expression in the nucellus. <line482> To define whether *ANT* expression in the nucellus was sufficient to control *PIN1*, we transformed wild type plants with a *pSPL/NZZ::ANT* construct. Even though we obtained 3 independent lines expressing the construct (Supplementary Figure 15A), we did not observe defect in the MMC development (Supplementary Figure15B, C). This result is not unexpected since, in *spl/nzz* mutants, several genes involved in the acquisition of chalaza identity are ectopically expressed. Therefore, we can conclude that *SPL/NZZ* is required to repress, in the nucellus, the genes required for chalaza specification and that, probably, *ANT* is only one of the genes that must be silenced to have a proper nucellus identity acquisition. Up to date, no information regarding ANT interacting partner and mode of action is available. <line490>

We believe that this topic could represents a new and exciting field of study that we, or potentially other research groups, will follow in the future.

Additionally, do spl-1 pSPL/NZZ::PIN1 and spl-1 pSPL/NZZ::ARF9 plants produce seeds? The plants are fertile and produce viable seeds. Despite this we did not perform a detailed analysis of seed development and plant fertility as, we believe, these are both developmental processes whose study goes beyond the scope of this work. This is a very interesting topics that we are planning to study in the near future.

“PCM” should be “PMC” in line 428, and “MCC” should be “MMC” in line 548.

We fixed the typos. Thank you for the comment.

REVIEWER COMMENTS

Reviewer #1 (Remarks to the Author):

Overall, the authors have addressed most of the previous concerns. However, a few minor issues remain:

1. The FLIM-FRET images in Figure 1 are of low quality, making them difficult to interpret. Please provide higher-resolution images with proper controls.

We thank the reviewer for the comment. The FRET-FLIM images presented in Figure 1 are generated by the SymPhoTime 64 software during FLIM measurement. In these images, each pixel is coloured according to its fluorescent signal decay time, as measured by the SymPhoTime 64 software. Therefore, the structures visualised in such images could be indeed of difficult interpretation. Therefore, based on reviewer 1 comment, we have included Figure 1D-H that show single-channel images showing GFP and mCHERRY signals in the different nuclei analysed during FLIM measurement. However, we believe that providing a visual representation of the change in the GFP lifetime in the different conditions (expression of SEP3-GFP and STK-GFP either alone or in combination with SPL/NZZ-mCHERRY) could be important. Therefore, we moved the images showing the GFP lifetime (highlighting the position of nuclei in the images) to the Supplementary Figure 3

2. While the AlphaFold prediction offers structural insights, it is computational and not experimentally validated. It would be more appropriate to move this figure to the supplemental section.

As suggested by the reviewer, we moved all the data concerning the Alpha Fold predictions of SPL/NZZ-SEPs-STK complexes to the Supplementary Figure 4.

3. Please indicate the sample size for each quantification and statistical analysis throughout the manuscript.

We ensured that information regarding the sample size of all quantifications and statistical analyses is properly provided in the legends or in the materials and methods section.

Reviewer #2 (Remarks to the Author):

Reviewer #3 (Remarks to the Author):

In this revised version of the manuscript, the authors addressed my main comments in detail.

1. They added FRET-FLIM data from tobacco experiments demonstrating the principal ability of direct interaction between SPL and SEP3 or SPL and STK in a plant cell. A test in Arabidopsis would have been preferable but this may have been technically challenging.

We thank the reviewer to considering the FRET-FLIM experiment in tobacco sufficient to prove the direct interaction of SPL, STK and SEP3 *in planta*.

2. In my opinion, the author's response to my concern about the temporal sequence of events regarding the down regulation of PIN1 in *spl-1* has been inadequate. My reasonings are as follows:

- I am confused about how the authors use the terms "identity acquisition/specification" and "differentiation," as these terms refer to two fundamentally different processes. I am also unclear about the exact roles that the authors propose for SPL and PIN1 in these processes. In the abstract, the authors mention that SPL is a principal regulator of MMC formation because its mutant phenotype affects MMC differentiation. Later, they continue to highlight the role of a regulatory network involving SPL/PIN1 in controlling MMC specification, yet they also discuss the role of SPL/PIN1 in MMC differentiation. I suppose the confusion stems from the question of whether the SPL/ANT/PIN1 axis is involved in specifying the MMC or its differentiation following specification, or in both processes. What is the authors's notion? Please clarify.

We thank the reviewer, and we agree that we must use more precisely the words 'specification' and 'differentiation'; therefore, we have corrected the text in accordance with the previously published manuscripts and our results.

SPL action has been previously associated with the MMC differentiation. Indeed, several works such as Wei-Cai Yang 1999 and Wei 2015, describe that *sp/* mutants do form an archesporium, which does

not differentiate into the MMC. Therefore, we agree to use the same terminology used in the previously published manuscript.

Our genetic data, along with previous studies, have shown that both the *ant* mutation (<https://doi.org/10.1242/dev.127.19.4227>) and the *pSPL::PIN1* construct can rescue the *spl* mutant phenotype; therefore, we have adopted the same terminology throughout the text as previously mentioned, following the suggestion of reviewer 3.

- At the onset of stage 2-I, a single large MMC is clearly visible in the wild type. Once the single MMC is enlarged further MMC development takes place during subsequent stages. The authors agree that the specification of a single MMC must occur before stage 2-I. The single large MMC is missing in *spl-1*. Based on cell morphology, the argument that MMC-like precursor cells appear during stages 0-III or 1-I in both the wild type and *spl-1* mutant is not convincing (Supplemental Figure 16). The larger cells present in the prospective nucellar region during stages 0/1 may or may not be MMC precursors. They could be cells that are dividing but have not yet undergone cytokinesis.

In any case, if SPL critically affects PIN-1 expression during MMC specification, it must also occur before stage 2-I. Otherwise, MMC specification depends on other SPL-regulated factors, and SPL/ANT-regulated PIN1 expression is relevant for MMC differentiation following specification. Do you suggest that a single MMC is specified in *spl-1* but does not enlarge and continue development?

Yes, as previously mentioned, we have agreed that, as described before, SPL is required for MMC differentiation rather than specification.

Failure in MMC enlargement/further differentiation could be explained by a role for SPL/ANT/PIN1/auxin in growth control (via for example auxin-mediated cell wall changes) and/or polarity control in the MMC. If so the text should clarify that specification of the single MMC is not apparently affected by SPL.

As we have written above in the response to the reviewer 3 's previous comment, we have agreed that the SPL-ANT-PIN1 action should be placed during the MMC differentiation phase, which occurs after MMC specification. We have changed the text accordantly. According to the timing of *PIN1* deregulation, we agree with the reviewer that the *PIN1* downregulation in *spl-1* is of pivotal importance to MMC differentiation and that the acquisition of MMC identity likely occurs at stage 2-I or even before. We properly described in the text the distinction in the SPL/ANT/PIN1 involvement in MMC differentiation with respect to the previous stages (line 502-521).

- There are also conflicting reports in the literature. The Colombo lab published data revealing that ectopic expression of SPL result in supernumerary MMCs (Mendes et.al. 2020 Development 147:dev194274). At the same time, other previous papers from the Colombo lab have provided evidence that, upon amiRNA-based downregulation of PIN1 in young ovules, in the *pin1-5* mutant, or in a cytokinin receptor mutant carrying ovules with no detectable PIN1pro:PIN1-GFP expression, early development of the embryo sac is blocked in the vast majority of ovules (Bencivenga et al., 2012, Plant Cell, 24:2886; Ceccato et al., 2013, PLoS ONE, 8:e66148). Thus, these data do not support a role for PIN1 in MMC specification. Additionally, under such conditions, MMC differentiation and meiosis would not be impaired to the extent that meiosis and megaspore formation could not be completed. How can these findings be reconciled with a model in c control MMC specification?

The model in which SPL/ANT/PIN1 control MMC differentiation summarised genetic data described in our manuscript and in previously published ones. We could complement the *spl* lack of a differentiated MMC by introducing *PIN1* under the *SPL* promoter. We obtained similar results also using the *pSPL::ARF9* construct. By contrast, expressing *pSPL::SHY2-6* in wild type ovules impaired MMC differentiation significantly, even if both *SPL* and PIN1 resulted still expressed. Therefore, we are confident that the differentiation of MMC required a proper auxin signalling response. The role of *ANT* in this process is based mainly on genetic data provided by previous works (for instance, the publication <https://doi.org/10.1242/dev.127.19.4227>) and on the role of *ANT* in repressing *PIN1*.

Regarding MMC differentiation in the mutant backgrounds cited by the reviewer, it is worth noting that *pin1-5* is a weak *pin1* allele. In this mutant allele, listils are formed and contain ovules, which are completely absent in *pin1* knock-out mutants. This suggests that a residual function of PIN1 is still present in *pin1-5* plants, which could be sufficient to support MMC differentiation. Concerning the analysis of the *amiRNA* knock-down line of *PIN1*, as previously shown, those plants present a reduction of *PIN1* expression of around 50% with respect to a wild-type situation, implying that *PIN1* could still be functional during MMC development. Nevertheless, even if the focus on these two manuscripts was to study the overall ovule development, as reported in Bencivenga et al., 2012, around 10 % of *pin1-5* ovule primordia were arrested as finger-like structures in which a clear MMC was not visible. The same was reported for *cre1-12 ahk2-2 ahk3-3* ovules (comparison of figure 2c and 2e Bencivenga et al 2012).

Minor point:

There are numerous typos scattered throughout the manuscript. Please correct them.

We have corrected the typos throughout the manuscript

Reviewer #4 (Remarks to the Author):

This resubmission includes a substantial amount of new data and revisions that address the reviewer's main concerns, particularly the genetic evidence supporting the role of SEP genes in MMC formation. The new result shows that the ectopic expression of ANT in the nucellus does not affect MMC formation. The authors propose that ANT functions redundantly with other genes to specify the nucellus identity. An earlier study showed that loss-of-function of ANT only partially restored MMC formation in *spl* mutant, and these MMCs failed to develop into embryo sacs. Thus, it is possible that ANT is not the sole direct target of SPL in inhibiting PIN1 activity.

The working models in Figure 7 are somewhat misleading. Furthermore, the authors should discuss how ANT and other genes controlling nucellus identity, together with other target genes contribute to MMC differentiation. Are pSPL/NZZ::ANT plants capable of producing functional embryo sacs or achieving normal seed set?

Thank you very much for the comment. We totally agree with reviewer 4 about the model presented in Figure 7. We apologise for the inaccuracy of the model in the previous manuscript version.

It has been reported that the *ant* mutation partially rescued the MMC differentiation defect in *spl* mutants (<https://doi.org/10.1242/dev.127.19.4227>); however, our experiments indicated that ANT is likely not the only direct SPL target involved in nucellus and MMC differentiation. Indeed, when we ectopically expressed *ANT* in the nucellus, we did not observe any issues with MMC differentiation. Therefore, there are other factors, not expressed in the nucellus in wild type, which remain unknown to date, that could work together or in parallel with ANT to impair MMC differentiation when expressed in the nucellus (as in the *spl* mutant). According to our data, the SPL-dependent repression of such factors should contribute to the establishment of the nucellar auxin response, which is central for MMC differentiation. The repression of chalazal genes could act on the spatiotemporal expression of PIN1. Indeed, besides ANT, BEL1 was also reported to be involved in PIN1 repression. Concerning BEL1, no information regarding its direct target genes is available to date. We discussed this point in the discussion section. <line 486-498>.

Regarding *pSPL/NZZ::ANT* plants, the majority of the T1 independent lines analysed successfully produced seeds, even if we noticed a reduction in the final seed production.